mechanical engineering

pump device, slanted axial-flow pump, flow loss, entropy production, numerical simulation

**Author for correspondence:**
Fan Yang
e-mail: fanyang@yzu.edu.cn

# Analysis of flow loss characteristics of slanted axial-flow pump device based on entropy production theory

Fan Yang[1,2], Zhongbin Li[1], Wenzhu Hu[1], Chao Liu[1,3], Dongjin Jiang[1], Dongsheng Liu[4] and Ahmed Nasr[1,5]

[1]College of Hydraulic Science and Engineering, Yangzhou University, Yangzhou 225009, People's Republic of China
[2]Key Laboratory of Fluid and Power Machinery, Ministry of Education, Chengdu 610039, People's Republic of China
[3]Hydrodynamic Engineering Laboratory of Jiangsu Province, Yangzhou 225009, People's Republic of China
[4]Nanjing Hydraulic Research Institute, Nanjing 210029, People's Republic of China
[5]Egyptian Ministry of Water Resources and Irrigation, Imbaba, Giza 12666, Egypt

FY, 0000-0001-5109-1772

Slanted axial-flow pump devices are widely applied in urban water supply, irrigation and drainage engineering fields. The second law of thermodynamics is applied to investigate the flow loss characteristics of the 30° slanted axial-flow pump model according to the flow loss analysis method of entropy production theory, so that the hydraulic loss characteristics can be revealed in internal flow process of the slanted axial-flow pump. The three-dimensional numerical simulation of the whole flow conduit in slanted axial-flow pump was conducted and the entropy production increased in the flow process was calculated. The location and distribution characteristics of the flow loss of the pump were qualitatively analysed. The results show that the entropy production in impeller is the highest among the pump components. With the increase of flow rate, the proportion of the entropy production in impeller in total value of the pump device increases continuously. The wall entropy production of impeller, guide vane and outlet conduit are lower than the mainstream entropy production, and the mainstream entropy production occupies the dominant position. As the flow rate grows, the proportion of turbulent dissipation entropy production decreases, and the proportion of wall dissipation entropy production increases. At $0.8Q_{bep}$, the proportion of turbulent dissipation entropy production is close to 74%,

which is about 2.8 times that of wall entropy production. Under $1.2Q_{bep}$ condition, the proportion of turbulent dissipation entropy production is just 5.5% higher than that of wall dissipation entropy production.

# 1. Introduction

The slanted axial-flow pump has the characteristics of small excavation depth, stable flow pattern and high efficiency, which is widely used in the Yangtze River Delta and the Pearl River Delta in China. Under ultra-low head conditions, the slanted axial-flow pump device shows good hydraulic performance, so it is widely used in urban drainage, plain agricultural irrigation and drainage fields (table 1).

With the increasing demand for water conveyance projects and irrigation and drainage, higher requirements are put forward to further optimize the performance of the slanted axial-flow pump. A lot of researches have been done on the internal flow field and flow characteristics of slanted axial-flow pump. Wang et al. [1] conducted numerical analysis on the whole flow conduit of large 15° slanted axial-flow pump, and the flow characteristics and working characteristics of the pump under zero-head condition were analysed. The comparison between calculated and measured results indicates that computational fluid dynamics (CFD) analysis has the ability to predict many ways of characteristics in pump accurately. Yang et al. [2] conducted numerical simulation on the 15° slanted axial-flow pump, analysed inlet conduit's hydraulic performance of the pump device and the distribution law of the flow force on the impeller blades. The prediction performance of the pump device was consistent with the model test performance. Zhang & Chen [3] investigated the cavitation flow of slanted axial-flow pump device by numerical simulation, with emphasis on estimating the hydraulic performance and pump device cavitation performance under various working conditions, and analysed the cavitation flow field by using the boundary vorticity flux. After studying the impact load and transient fluid pressure of the axial-flow pumping station system under non-adjustable conditions through model tests and numerical simulations. Fu et al. [4] indicate the change of flow pattern and blade pressure distribution in the impeller area during the transient process of the axial-flow pump start-up. Kan et al. [5–7] used CFD technology to analyse the typical overflow structure and pressure fluctuation characteristics of axial-flow pump, and carried out comparative analysis under design conditions and stall conditions. And the low-frequency pressure fluctuation mechanism of pump unit under stall condition was revealed. Wang et al. [8] investigated the flow deviation phenomenon in S-shaped flow passage in a slanted axial-flow pumping station, the 'unwinding' flow structure was discovered and the development of 'unwinding' structure in different locations of S-shaped discharge passage was revealed.

The traditional method of estimating hydraulic loss is to calculate the pressure drop on the inlet and outlet to obtain the relatively general hydraulic loss in the conduit, while the detailed distribution and size of hydraulic loss at different positions in the conduit cannot be known. The entropy production theory has obvious advantages in the evaluation of hydraulic loss, which can accurately locate the source of hydraulic loss, and quantitatively and qualitatively analyse the size and distribution of hydraulic loss. Therefore, it can provide a theoretical basis for the improvement and optimization of components with large hydraulic loss. Scholars introduced the entropy production theory to solve and analyse the entropy production rate of rotating machinery, and linked the energy loss with the entropy production [9–13]. Entropy production can directly reflect the position of irreversible loss in the fluid and the distribution of energy dissipation, which provides a new method for pump performance improvement and hydraulic optimization. Herwig & Kock [14] discussed direct and indirect methods in detail, which can provide entropy generation information with different accuracy. Hou et al. [15] introduced the local entropy production analysis method to assess a centrifugal water pump's energy loss. The results show that wall entropy production and turbulent entropy production accounts for a large portion of the whole entropy production in pump. And the entropy production distribution in impeller domain is discussed. Li et al. [16] applied the entropy production theory to systematically analyse the hump characteristics and hysteresis effect of pump-turbine under pump conditions. The hump characteristics and the accompanying hysteresis phenomenon are caused by the backflow at the inlet of the runner and the separation vortex near the hub and trailing edge, which is related to the flow direction. Pei et al. [17] evaluate the influence of the distance between the guide vane and the impeller on pump device's flow loss distribution by entropy production theory. The results draw that the

**Table 1.** Nomenclature.

| | |
|---|---|
| $Q_{bep}$ | flow rate of best efficiency point $(m^3 \cdot s^{-1})$ |
| CFD | computational fluid dynamics |
| TLF | tip leakage flow |
| $D$ | diameter of impeller (m) |
| $n$ | rotational speed (r.p.m.) |
| GCI | grid convergence index |
| $y+$ | dimensionless wall distance |
| $\rho$ | density of water $(kg \cdot m^{-3})$ |
| $t$ | time (s) |
| $u$ | velocity $(m \cdot s^{-1})$ |
| $x$ | coordinate (m) |
| $p$ | pressure (Pa) |
| $\mu$ | dynamic viscosity $(Pa \cdot s)$ |
| $i, j$ | coordinate axis directions |
| $k$ | turbulent kinetic energy $(m^2 \cdot s^{-2})$ |
| $\varepsilon$ | turbulent kinetic energy dissipation rate $(m^2 \cdot s^{-3})$ |
| $\tau$ | shear stress $(N \cdot m^{-2})$ |
| $\mathbf{v}$ | velocity vector $(m \cdot s^{-1})$ |
| $\mu_t$ | turbulent dynamic viscosity $(Pa \cdot s)$ |
| $\dot{S}_{\bar{D}}'''$ | entropy production rate induced by average velocity $(W \cdot m^{-3} \cdot K^{-1})$ |
| $\dot{S}_{D'}'''$ | entropy production rate induced by fluctuating velocity $(W \cdot m^{-3} \cdot K^{-1})$ |
| $\mu_{eff}$ | effective dynamic viscosity $(Pa \cdot s)$ |
| Re | Reynolds number |
| $\dot{S}_{D}'''$ | mainstream entropy production rate $(W \cdot m^{-3} \cdot K^{-1})$ |
| $\dot{S}_{w}'''$ | wall entropy production rate $(W \cdot m^{-2} \cdot K^{-1})$ |
| $S_{pro,\bar{D}}$ | entropy production induced by time-averaged velocity $(W \cdot K^{-1})$ |
| $S_{pro,D'}$ | entropy production induced by fluctuating velocity $(W \cdot K^{-1})$ |
| $S_{pro,W}$ | entropy production at the wall $(W \cdot K^{-1})$ |
| $S_{pro}$ | total entropy production of flow field $(W \cdot K^{-1})$ |
| $h_{pro}$ | hydraulic loss calculated by entropy production (m) |
| $T$ | temperature of the flow field (K) |
| $\dot{m}$ | mass flow rate $(kg \cdot s^{-1})$ |
| $V$ | volume $(m^3)$ |
| $A$ | area $(m^2)$ |
| $E_\eta$ | total uncertainty of efficiency (%) |
| $E_{\eta,S}$ | systematic uncertainty of the experimental system (%) |
| $E_{\eta,R}$ | random uncertainty of the experimental system (%) |
| $h_p$ | hydraulic loss calculated by pressure drop (m) |
| $P_{in}$ | total pressure of conduit inlet (Pa) |
| $P_{out}$ | total pressure of conduit outlet (Pa) |
| $W_S$ | input power of impeller (W) |
| $R$ | ratio of hydraulic loss |

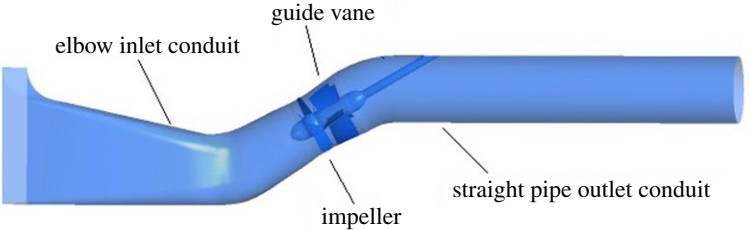

**Figure 1.** 30° slanted axial-flow pump device model.

**Table 2.** 30° slanted axial-flow pump device model parameters.

| parameter | value |
|---|---|
| impeller nominal diameter $D$ (m) | 1.68 |
| impeller blades number | 4 |
| designed flow rate $Q_{bep}$ (m³·s⁻¹) | 10.66 |
| rated speed $n$ (r.p.m.) | 250 |
| guide vane blades number | 5 |
| hub ratio | 0.33 |
| $nD$ | 420 |
| impeller blade place angle | 0° |

efficiency of the pump during forward operation is affected by the distance between the impeller and the guide vane of the bidirectional axial-flow pump, and main source of loss of the pump device is turbulence dissipation. Chang *et al.* [18] investigated the types, sizes and locations of hydraulic losses in novel self-priming pump under the blades with various thickness distributions based on entropy production theory. Ji *et al.* [19] applied the entropy production method to analyse the effect of the impeller tip clearance on internal flow field and hydraulic loss in a mixed flow pump. They concluded that the energy loss in the impeller domain may raise as the tip leakage flow (TLF) grows, while due to the existence of TLF, the energy loss in the guide vane domain is suppressed to a certain degree.

Flow loss in pump is mainly the internal flow loss induced by fluid motion. Many scholars [20–24] have studied the connection between the flow loss and hydraulic performance of centrifugal pump, mixed flow pump and other types of pumps in detail. However, the research on the flow loss of slanted axial-flow pump under different working conditions is not enough. There is little open literature on the specific location and loss mechanism of internal flow loss in a slanted axial-flow pump. In this paper, the entropy production theory in view of numerical results is applied to calculate the entropy production increased in internal flow process of 30° slanted axial-flow pump device, and the distribution law of the flow loss of pump device is analysed.

## 2. Simulation model

### 2.1. Three-dimensional pump modelling

The 30° slanted axial-flow pump device includes elbow inlet conduit, impeller, guide vane and straight pipe outlet conduit. In order to reduce backflow in the inlet conduit and the flow pattern effect on the impeller, an open inlet extension section is set up before the elbow inlet conduit. The length of the inlet extension section is equal to the length of the elbow inlet conduit, and the height of inlet extension section is 1.2 times the elbow inlet conduit. A three-dimensional pump model is established for the calculation domain of elbow inlet conduit, straight pipe outlet conduit, impeller and guide vane. The impeller nominal diameter $D$ is 1.68 m, the impeller blades number is 4, the tip clearance is 0.2 mm, the guide vane blades number is 5, the hub ratio is 0.33, the rotating speed $n$ is 250 r.p.m., the $nD$ value is 420 and $Q_{bep}$ is 10.66 m³ s⁻¹. The main parameters of the 30° slanted axial-flow pump are shown in table 2. Figure 1

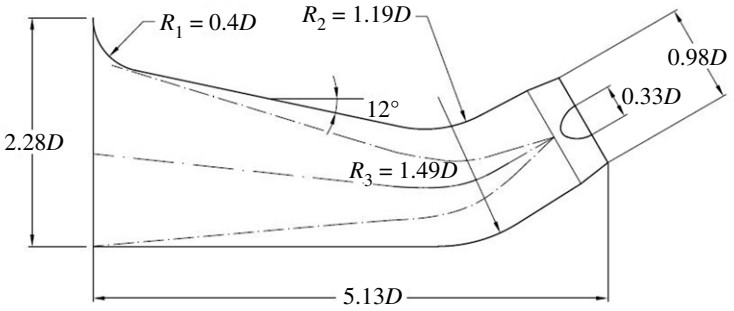

**Figure 2.** Geometric single-line diagram of elbow inlet conduit.

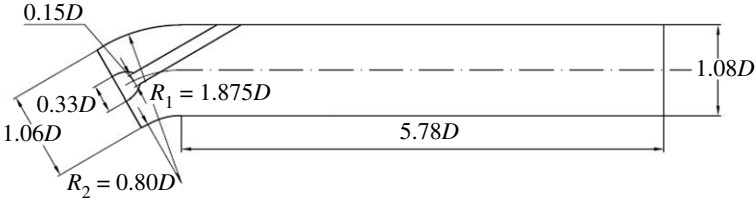

**Figure 3.** Geometric single-line diagram of straight pipe outlet conduit.

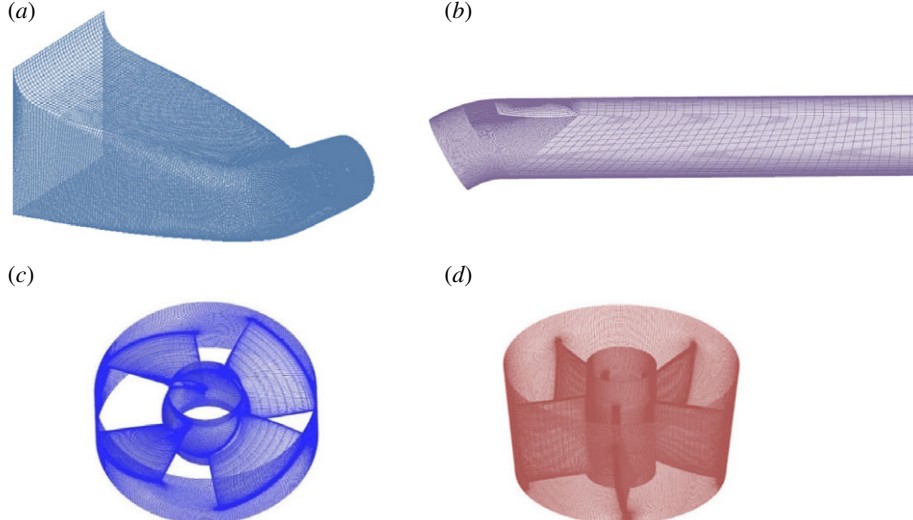

**Figure 4.** Meshing of pump device components. (*a*) Elbow inlet conduit. (*b*) Straight pipe outlet conduit. (*c*) Impeller. (*d*) Guide vane.

shows the three-dimensional whole flow conduit of pump model, and the geometric single-line diagrams of the elbow inlet conduit and straight pipe outlet conduit are shown in figures 2 and 3.

## 2.2. Mesh generation

The structured grids are employed to mesh the parts of the 30° slanted axial-flow pump device. The structured grid subdivision results of inlet extension section, inlet conduit, outlet conduit, impeller calculation domain and guide vane calculation domain are shown in figure 4. The grid quality is tested by angle value and Jacobi determinant, and the Jacobi determinant values are greater than 0.4. The grid orthogonality of impeller is among 28°–155°, and that of guide vane is among 31°–156°, which meets the requirements of grid orthogonality between 15° and 165°. Scalable wall function is used to simulate the near solid wall flow of the pump device. The $y^+$ value is the dimensionless distance between the first grid node and the wall. In the design flow conditions, the $y^+$ value of each flow passage component in 30° axial-flow pump device is as follows: the $y^+$ value in elbow inlet conduit is about 289, the $y^+$ value in impeller is about 28, the $y^+$ value in guide vane is about 53, and the $y^+$ value in straight pipe outlet conduit is about 201. The $y^+$ value meets the requirements of numerical simulation [25].

**Table 3.** Boundary conditions setting.

| boundary conditions | |
|---|---|
| locations | type |
| model pump inlet | mass flow rate |
| model pump outlet | static pressure, 1 standard atmosphere |
| solid wall surfaces | no-slip wall |
| interface on both sides of impeller | stage |
| static and static interface | none |
| convergence criterion | $10^{-5}$ |

## 2.3. Boundary condition

ANSYS CFX is applied to simulate the steady incompressible flow in 30° slanted axial-flow pump device. The mass flow rate is set as the boundary condition at the inlet. The outlet of the straight pipe outlet conduit is set as the static pressure with a standard atmospheric pressure. The solid wall is set to a no-slip boundary condition, that is, the normal and tangential velocities of fluid particles on the wall are zero.

The calculation domain of the pump device in this work includes five sub-calculation domains, namely, the inlet extension section, the elbow inlet conduit, the impeller domain, the guide vane domain and the straight pipe outlet conduit. The data transmission at the interface of the adjacent calculation domains is involved in the simulation process. The interface is connected by the general grid interface (GGI) method, which allows the grid types on both sides of the interface to be different, and the nodes are not one-to-one correspondence, which has strong adaptability. The impeller calculation domain is a rotating calculation domain. The interfaces on both sides of impeller calculation domain are dynamic and static interfaces and set as Stage. Stage model refers to the mixing plane model (MPM). MPM performs circumferential average of physical quantities (pressure, velocity, etc.) on the mixing plane between the rotating domain and the stationary domain. And the distribution of physical quantities is introduced into the adjacent computational domain as boundary conditions. With the iteration of the calculation, the distribution of physical quantities on the adjacent boundaries of the mixing plane tends to be consistent until the calculation converges. MPM is mostly applied in hydraulic machinery interface such as axial-flow pump and axial-flow turbine. None model is suitable for the domains with no frame change or pitch change such as the static interface of guide vane domain and straight pipe outlet conduit domain. The convergence residual is set to $10^{-5}$. The boundary conditions are shown in table 3.

## 2.4. Mesh independence and convergence analysis

The grid independence and convergence analysis of the whole pump device are carried out under the optimal conditions. Table 4 shows the pump device efficiency of different grid numbers under the same boundary conditions and controlling equation. It is clear that the efficiency of the pump device increases as the grids number grows and when the grids number comes to 4.83 million, the pump device efficiency changes within 0.3%.

The grid quality, grid type and method have an impact on the calculation speed and the accuracy of the results in numerical simulation. The grid convergence is analysed to verify the effect of different grid size on the simulation results. Roache [26] proposed using grid convergence index (GCI) to calculate the discrete error and judge the grid convergence. Nanda Kumar & Govardhan [27] and Sedrez *et al.* [28] used GCI to verify grid independence. The procedure for calculating discrete errors in this paper is based on Celik *et al.* [29]. Three groups of grids with small change in the efficiency of the pump device are selected for numerical simulation under the optimal condition. The three groups of grids were $N_1 = 5.14$ million, $N_2 = 4.83$ million and $N_3 = 4.29$ million, respectively. Table 5 is the calculation results with the reference of [29]. $GCI_{21}$ and $GCI_{32}$ are less than 1%, indicating that the discrete error is within a reasonable range [27]. The three groups of grids are suitable for numerical simulation.

**Table 4.** Grid number independence verification.

| number | grid number ($\times 10^6$) | efficiency (%) | error (%) |
|---|---|---|---|
| 1 | 1.64 | 74.86 | |
| 2 | 1.98 | 75.20 | 0.34 |
| 3 | 2.62 | 75.90 | 0.70 |
| 4 | 3.01 | 76.73 | 0.83 |
| 5 | 3.53 | 77.69 | 0.96 |
| 6 | 4.29 | 78.32 | 0.63 |
| 7 | 4.83 | 78.55 | 0.23 |
| 8 | 5.14 | 78.50 | −0.05 |

**Table 5.** Calculation for discretization error.

| parameter | calculation results |
|---|---|
| $N_1$, $N_2$, $N_3$ | 5 140 000, 4 830 000, 4 290 000 |
| $r_{21}$, $r_{32}$ | 1.021, 1.041 |
| $\phi_1$, $\phi_2$, $\phi_3$ | 78.50%, 78.55%, 78.32% |
| $p$ | 74.12 |
| $\phi_{ext}^{21}$, $\phi_{ext}^{32}$ | 0.785, 0.786 |
| $e_a^{21}$, $e_a^{32}$ | 0.0006, 0.0029 |
| $e_{ext}^{21}$, $e_{ext}^{32}$ | 0.0002, 0.00017 |
| $GCI_{21}$, $GCI_{32}$ | 0.02%, 0.02% |

**Table 6.** Grid number of each component.

| component | elbow inlet conduit | impeller | guide vane | straight pipe outlet conduit | total |
|---|---|---|---|---|---|
| grid number ($\times 10^6$) | 1.74 | 0.96 | 1.22 | 0.91 | 4.83 |

Considering the higher efficiency of the pump device at 4.83 million grids, 4.83 million is selected for final mesh generation. The grid number of each component is shown in table 6.

## 2.5. Control equation and turbulence model

The fluid inside the pump device can be approximately regarded as incompressible three-dimensional viscous turbulence. The continuity equation and momentum equation can be expressed as equations (2.1) and (2.2). The control equation is discretized by control volume finite-element method (CVFEM). The discrete equation is solved by the fully implicit coupled algebraic multi-grid method. In the discretization process, the convection term adopts the high-resolution scheme, and the other terms adopt the central difference scheme.

$$\frac{\partial(u_i)}{\partial(x_i)} = 0 \tag{2.1}$$

and

$$\frac{\partial(\rho u_i)}{\partial t} + \frac{\partial}{\partial x_j}(\rho u_i u_j) = -\frac{\partial p}{\partial x_i} + \frac{\partial}{\partial x_j}\left(\mu\left(\frac{\partial u_i}{\partial x_j} + \frac{\partial u_j}{\partial x_i}\right)\right) + S_i, \tag{2.2}$$

where $\rho$ is water density, $t$ is time, $u$ is velocity, $x$ is space coordinate, $p$ is pressure, $\mu$ is fluid viscosity coefficient, $S$ is external source term, $i$, $j$ are coordinate axis directions.

The RNG $k$–$\varepsilon$ turbulence model is based on the renormalization group analysis of the N–S equation. Compared with the standard $k$–$\varepsilon$ turbulence model, the model constants of the turbulent kinetic energy dissipation rate $\varepsilon$ transport equation is different, and the calculation accuracy of the RNG $k$–$\varepsilon$ turbulence model for the turbulent dissipation term in the flow separation region is improved. This correction takes into account the rotation effect in the average flow and has advantages in dealing with large curvature, strong rotation and high strain rate flow in the pump impeller [30–33]. Therefore, the RNG $k$–$\varepsilon$ model is applied to simulate the 30° slanted axial-flow pump

$$\rho \frac{\partial k}{\partial t} + \rho \frac{\partial (ku_i)}{\partial x_i} = \frac{\partial}{\partial x_j} \left[ \sigma_k (\mu + \mu_t) \frac{\partial k}{\partial x_j} \right] + \mu_t \left( \frac{\partial u_i}{\partial x_j} + \frac{\partial u_j}{\partial x_i} \right) \frac{\partial u_i}{\partial x_i} - \rho \varepsilon \tag{2.3}$$

and

$$\rho \frac{\partial (\varepsilon)}{\partial t} + \frac{\partial (\rho \varepsilon u_i)}{\partial x_i} = \frac{\partial}{\partial x_j} \left[ \sigma_\varepsilon (\mu + \mu_t) \frac{\partial \varepsilon}{\partial x_j} \right] + \frac{C_{1\varepsilon}}{k} \mu_t \left( \frac{\partial u_i}{\partial x_j} + \frac{\partial u_j}{\partial x_i} \right) \frac{\partial u_i}{\partial x_i} - C_{2\varepsilon} \rho \frac{\varepsilon^2}{k} - R, \tag{2.4}$$

where $R$, $\mu_t$ could be calculated by equations (2.5) and (2.7)

$$R = \frac{C_\mu \eta^3 (1 - \eta/\eta_0)}{1 + \beta \eta^3}, \tag{2.5}$$

$$\eta = \frac{k}{\varepsilon} \sqrt{\left( \frac{\partial u_i}{\partial x_j} + \frac{\partial u_j}{\partial x_i} \right) \frac{\partial u_i}{\partial x_j}} \tag{2.6}$$

and

$$\mu_t = \rho C_\mu \frac{k^2}{\varepsilon}, \tag{2.7}$$

where $k$ is the turbulent kinetic energy, $\varepsilon$ is the dissipation rate, $\mu$ is the dynamic viscosity coefficient, $\mu_t$ is turbulent dynamic viscosity, the constants are: $\eta_0 = 1.38$, $\beta = 0.012$, $C_\mu = 0.0845$, $C_{1\varepsilon} = 1.42$, $C_{2\varepsilon} = 1.68$, $\sigma_k = 0.72$ and $\sigma_\varepsilon = 0.75$.

# 3. Entropy production theory

In order to evaluate the energy loss distribution in the components of the 30° slanted axial-flow pump in detail, entropy production theory is introduced in this paper. In the view of second law of thermodynamics, entropy increase is always accompanied in an actual fluid system. In the process of pump impeller rotation and water flow, it could be considered that the water is constant in temperature. The viscous force in the boundary layer near the solid surface will convert kinetic and pressure energy of the fluid into internal energy and dissipate, which results in an increase in entropy production. Adverse flow patterns such as vortex and secondary reflux will lead to the raise of hydraulic loss, accompanying the increase in entropy production. Consequently, the entropy production theory is suitable for accurately evaluating internal energy dissipation in pump device.

In turbulent motion, the fluid velocity includes two parts: average velocity and fluctuating velocity, and the corresponding entropy production rate also includes two parts: one is the entropy production rate caused by the average velocity flow motion; the other is the entropy production rate caused by turbulent fluctuating velocity. The entropy production rate caused by average velocity can be calculated by equation (3.1).

$$\dot{S}'''_{\bar{D}} = \frac{2\mu_{\text{eff}}}{T} \left[ \left( \frac{\partial \overline{u_1}}{\partial x_1} \right)^2 + \left( \frac{\partial \overline{u_2}}{\partial x_2} \right)^2 + \left( \frac{\partial \overline{u_3}}{\partial x_3} \right)^2 \right] + \frac{\mu_{\text{eff}}}{T} \left[ \left( \frac{\partial \overline{u_2}}{\partial x_1} + \frac{\partial \overline{u_1}}{\partial x_2} \right)^2 + \left( \frac{\partial \overline{u_2}}{\partial x_3} + \frac{\partial \overline{u_3}}{\partial x_2} \right)^2 + \left( \frac{\partial \overline{u_3}}{\partial x_1} + \frac{\partial \overline{u_1}}{\partial x_3} \right)^2 \right]. \tag{3.1}$$

The entropy production rate of turbulent kinetic energy dissipation caused by fluctuating velocity is calculated by equation (3.2).

$$
\begin{aligned}
\dot{S}_{D'}''' = {} & \frac{2\mu_{\mathrm{eff}}}{T}\left[\left(\frac{\partial u'_1}{\partial x_1}\right)^2 + \left(\frac{\partial u'_2}{\partial x_2}\right)^2 + \left(\frac{\partial u'_3}{\partial x_3}\right)^2\right] \\
& + \frac{\mu_{\mathrm{eff}}}{T}\left[\left(\frac{\partial u'_2}{\partial x_1} + \frac{\partial u'_1}{\partial x_2}\right)^2 + \left(\frac{\partial u'_2}{\partial x_3} + \frac{\partial u'_3}{\partial x_2}\right)^2 + \left(\frac{\partial u'_3}{\partial x_1} + \frac{\partial u'_1}{\partial x_3}\right)^2\right]
\end{aligned}
\tag{3.2}
$$

and

$$
\mu_{\mathrm{eff}} = \mu + \mu_t, \tag{3.3}
$$

where $\mu_{\mathrm{eff}}$ is effective dynamic viscosity, $\mu$ is dynamic viscosity, $\mu_t$ is turbulent dynamic viscosity.

The mainstream entropy production rate induced by turbulent motion can be expressed as equation (3.4).

$$
\dot{S}_D''' = \dot{S}_{\bar{D}}''' + \dot{S}_{D'}'''. \tag{3.4}
$$

Since the entropy production rate caused by fluctuating velocity cannot be directly acquired, in the RNG $k$–$\varepsilon$ turbulent model, when Reynolds number Re → ∞, the entropy production rate caused by fluctuating velocity can be expressed as equation (3.5).

$$
\dot{S}_{D'}''' = \frac{\rho\varepsilon}{T}, \tag{3.5}
$$

where $\varepsilon$ is turbulent kinetic energy dissipation rate.

In the process of fluid transition from the turbulent core far away from the near wall to the laminar boundary layer, obvious velocity gradient will appear near the wall due to the influence of flow viscosity. Herwig & Kock [14] found that using DNS method to calculate entropy production without special treatment of wall entropy production would result in unacceptable error. Zhang *et al.* [34] and Duan *et al.* [35,36] proposed to calculate the entropy production rate in wall region by using the wall shear stress and the velocity near the wall. The entropy production rate in wall region can be calculated in equation (3.6). Scholars [15,16,18,19] verified the reliability of this method by comparing the hydraulic loss calculated by the entropy production and pressure drop.

$$
\dot{S}_w''' = \frac{\boldsymbol{\tau} \cdot \boldsymbol{v}}{T}, \tag{3.6}
$$

where $\boldsymbol{\tau}$ is wall shear stress, $\boldsymbol{v}$ is velocity vector of the first grid centre at the wall.

In order to obtain the total entropy production in the flow field, the mainstream entropy production rate and the wall entropy production rate could be integrated and calculated by equation (3.7).

$$
\left.
\begin{aligned}
S_{\mathrm{pro},\bar{D}} &= \int_v \dot{S}_{\bar{D}}''' \, \mathrm{d}V, \\
S_{\mathrm{pro},D'} &= \int_v \dot{S}_{D'}''' \, \mathrm{d}V, \\
S_{\mathrm{pro},W} &= \int_A \frac{\boldsymbol{\tau} \cdot \boldsymbol{v}}{T} \, \mathrm{d}A \\
S_{\mathrm{pro}} &= S_{\mathrm{pro},\bar{D}} + S_{\mathrm{pro},D'} + S_{\mathrm{pro},W},
\end{aligned}
\right\}
\tag{3.7}
$$

and

where $S_{\mathrm{pro},\bar{D}}$ is entropy production caused by time-averaged velocity, $S_{\mathrm{pro},D'}$ is entropy production caused by fluctuating velocity, $S_{\mathrm{pro},W}$ is entropy production at the wall, $S_{\mathrm{pro}}$ is total entropy production of flow field, $V$ is the volume of fluid domain, $A$ is wall area of the fluid domain.

Assuming that the temperature of the flow field of the pump remains unchanged during operation, that is, the temperature change of the flow field is ignored, the hydraulic loss calculated due to the increase of entropy production in the flow field can be expressed as equation (3.8) [16].

$$
h_{\mathrm{pro}} = \frac{T \cdot S_{\mathrm{pro}}}{\dot{m}g}, \tag{3.8}
$$

where $h_{\mathrm{pro}}$ is the hydraulic loss calculated by entropy production, $T$ is the temperature of the flow field, $m$ is the mass flow rate of the pump device.

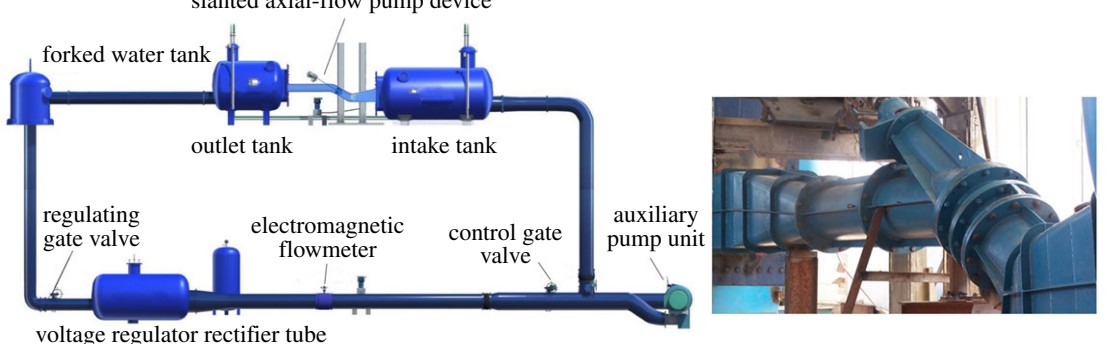

**Figure 5.** Schematic diagram of high-precision test bench.

**Table 7.** Type and system error of test equipment.

| term | equipment | type | system error (%) |
|---|---|---|---|
| flow rate | electromagnetic flowmeter | E-mag DN400 | ±0.18 |
| head | differential pressure transmitter | EJA110A | ±0.20 |
| torque | digital torque and speed sensor | TS-3100B | ±0.20 |
| rotation speed | digital torque and speed sensor | TS-3100B | ±0.05 |

# 4. Experimental device and uncertainty analysis

The 30° slanted axial-flow pump device's energy performance test is conducted on the high-precision closed-loop test bench in the key laboratory of Jiangsu Province. Figure 5 is the schematic diagram of the test bench. The test bench mainly consists of intake tank, pressure outlet tank, electromagnetic flowmeter, control valve, torque meter, etc. The main monitoring quantities in the experiment include flow rate, static pressure of pump device inlet and outlet, and torque and speed of pump shaft. The equipment for monitoring the test quantities is shown in table 7. The model impeller diameter is 300 mm, the rotational speed is 1400 r.p.m., the impeller blades number is 4, and the blades placement angle is 0°. The guide blades number is 5.

The experimental equipment has been verified by the metrological calibration department approved by the state. The system error parameters of the main test equipment are shown in table 7. The uncertainty of efficiency in model test rig pump device is calculated by the reference [37].

The uncertainty of the system is determined by each test equipment's systematic error and the previous test results. The overall random uncertainty is based on the experimental data of the pump device system, and the probability and statistics method is used to calculated the random uncertainty. The uncertainty of the experimental system $E_{\eta,S}$ is 0.339%, the total random uncertainty $E_{\eta,R}$ is 0.156%, and the test bench comprehensive uncertainty $E_{\eta}$ is 0.373%, which is within the requirements of the water conservancy industry standard of the People's Republic of China *Code of Practice for Model Pump and Its Installation Acceptance Tests* (SL140-2006).

# 5. Results and discussion

## 5.1. Verification of numerical simulation

The geometry of the prototype pump device is converted to the physical model according to the principle of equal $nD$ value to validate the numerical simulation reliability. The converted physical model pump device's impeller rotating speed is 1400 r.p.m., impeller nominal diameter is 0.3 m, and the prototype of the flow components are scaled to the model by 0.1786. The head and efficiency of the physical pump device model under different flow conditions are collected with the 0° blade placement angle. The equivalent efficiency conversion method is used to convert the physical model test results of the pump device to the prototype. Figure 6 shows the comparison between the performance

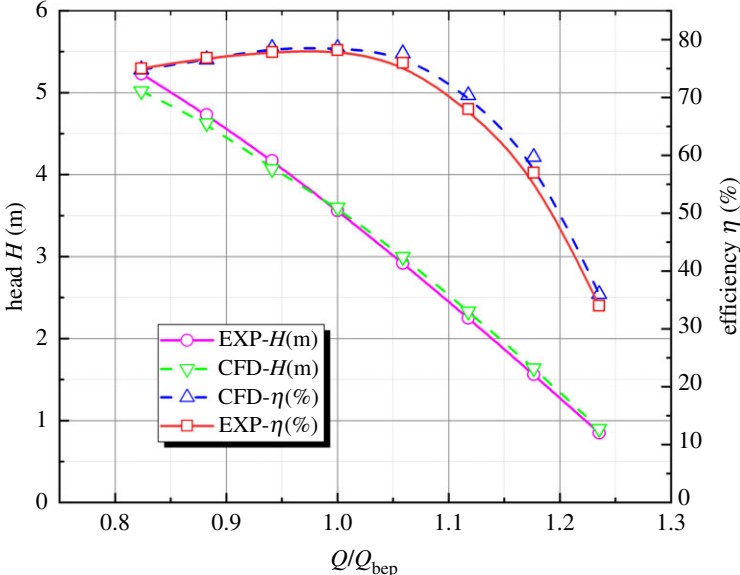

**Figure 6.** Pump performance characteristics of model test and numerical simulation.

characteristics of the converted physical model and the numerical simulation. The variation trend of the predicted performance curve and the test curve of the 30° slanted axial-flow pump device model is basically the same. The maximum difference between the numerical predicted pump device head and the physical model test value is 4.0%, and the minimum difference is 1.2%. The numerical simulation head is slightly lower than the model test when the flow rate is small. The pump head is well fitted near $1.0Q_{bep}$, and the numerical simulation head is a little bit higher than model test at large flow rate. The maximum difference between pump device efficiency simulated by numerical simulation and efficiency of the physical model test is 5.88%, and the minimum difference is 0.3%. At small flow rate and $1.0Q_{bep}$, the numerical simulation flow–efficiency characteristic curve is basically in agreement with the test of physical model, which indicates the effectiveness and reliability of numerical simulation of 30° slanted axial-flow pump device.

## 5.2. Total entropy production at different flow rates of each component

The entropy production of 30° slanted axial-flow pump device at various flow rates is calculated. Figure 7 shows the entropy production of each flow component under different flow conditions. The entropy production proportion in each flow component to the whole pump device under different flow conditions is shown in figure 8.

The impeller domain's entropy production is the highest in all pump components, reaching more than $180\,W\,K^{-1}$, and the entropy production grows gradually with an increased flow rate. The entropy production proportion of impeller in total pump device entropy production increases continuously. The entropy production in straight pipe outlet conduit is large under low flow rate condition, reaching nearly $200\,W\,K^{-1}$, which has a relationship with the disorder flow pattern in outlet conduit at small flow rate condition. When the flow rate increases continuously, the entropy production in straight pipe outlet decreases sharply. When the flow rate is greater than the design flow rate, the decrease trend of entropy production gradually slows down and entropy production proportion in outlet conduit decreases. Entropy production change trend in guide vane domain is similar to that of straight pipe outlet conduit, but the entropy production is smaller as a whole. The entropy production in guide vane domain reduces with an increased flow rate, and the trend gradually slows down, indicating that the effect of recovery kinetic energy of guide vane enhances. The elbow inlet conduit's entropy production increases as the flow rate grows; however, the value of entropy production is less than $0.1\,W\,K^{-1}$, which accounts for a very small proportion in whole flow conduit's total entropy production.

The total entropy production in whole pump device system consists of direct dissipation entropy production, turbulent dissipation entropy production and wall dissipation entropy production. Table 8 is the distribution of three types of entropy production ratio of whole flow conduit in 30° slanted

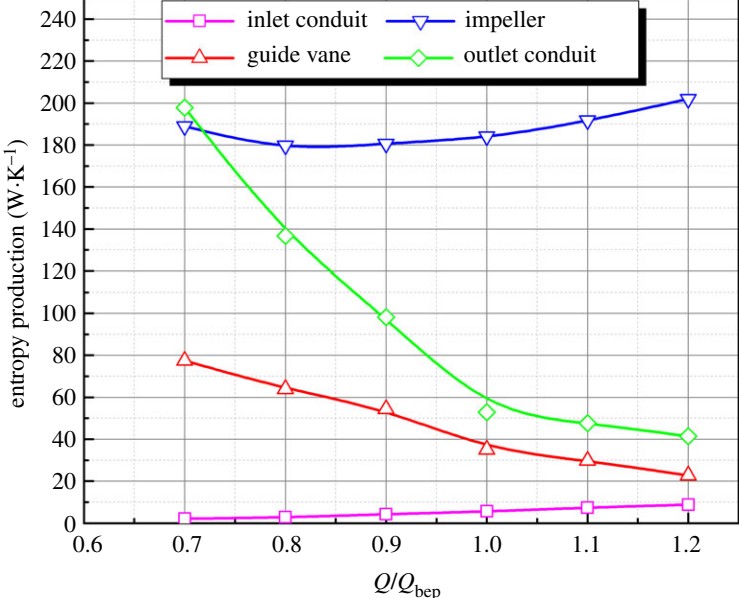

**Figure 7.** Entropy production of components at different flow rates.

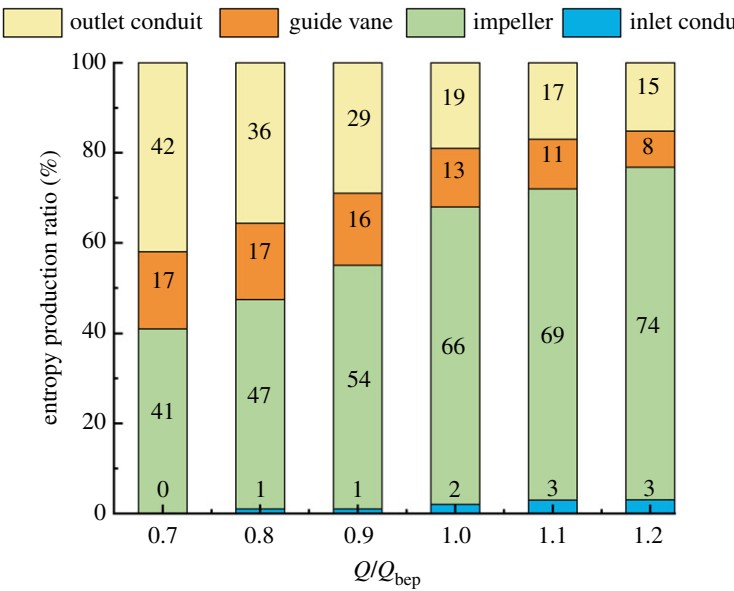

**Figure 8.** Proportion of entropy production of components at different flow rates.

axial-flow pump device at various flow rate conditions. Under various flow rate conditions, the proportion of turbulent dissipation entropy production and wall dissipation entropy production in total entropy production is way above the direct dissipation entropy production. As the flow rate grows, turbulent dissipation entropy production proportion reduces, and wall dissipation entropy production proportion increases. At $0.8Q_{\text{bep}}$ flow condition, the proportion of turbulent dissipation entropy production is close to 74%, which is about 2.8 times the wall dissipation entropy production. Under $1.2Q_{\text{bep}}$ flow condition, the proportion of turbulent dissipation entropy production is just 5.55% higher than that of wall dissipation entropy production.

## 5.3. Validation of hydraulic loss calculated by entropy production

The traditional hydraulic loss calculation method of hydraulic machinery is usually obtained by calculating the total pressure drop between the inlet and outlet. For elbow inlet conduit, guide vane and straight pipe outlet conduit, the hydraulic loss is calculated by equation (5.1). The

**Table 8.** Three types of entropy production proportion.

| flow rate | proportion of direct dissipation entropy production (%) | proportion of turbulent dissipation entropy production (%) | proportion of wall dissipation entropy production (%) |
|---|---|---|---|
| $0.7Q_{bep}$ | 0.0013 | 72.73 | 27.27 |
| $0.8Q_{bep}$ | 0.0016 | 73.58 | 26.42 |
| $0.9Q_{bep}$ | 0.0020 | 60.69 | 39.31 |
| $1.0Q_{bep}$ | 0.0025 | 57.02 | 42.97 |
| $1.1Q_{bep}$ | 0.0026 | 54.92 | 45.08 |
| $1.2Q_{bep}$ | 0.0026 | 52.79 | 47.21 |

impeller domain belongs to the rotating part, and the hydraulic loss is the work obtained by subtracting the total water pressure rise from the total input power of the impeller domain, and is calculated by equation (5.2).

$$h_p = \frac{P_{in} - P_{out}}{\rho g} = \frac{\left[(1/A)\sum_{i=1}^{n} p_i |A_i|\right]_{in} - \left[(1/A)\sum_{i=1}^{n} p_i |A_i|\right]_{out}}{\rho g} \tag{5.1}$$

and

$$h_p = \frac{W_s}{\rho g Q} - \frac{\left[(1/A)\sum_{i=1}^{n} p_i |A_i|\right]_{in} - \left[(1/A)\sum_{i=1}^{n} p_i |A_i|\right]_{out}}{\rho g}, \tag{5.2}$$

where $h_p$ is the hydraulic loss calculated by pressure drop, $P_{in}$ is the total pressure of the inlet, $P_{out}$ is the total pressure of the outlet, $A_i$ is the area of $i$th grid, $P_i$ is the pressure of $i$th grid, $W_S$ is the input power of impeller, $Q$ is the flow rate, $g$ is the local gravity acceleration.

In order to verify the accuracy of hydraulic loss calculated by entropy production theory, the hydraulic loss ratio $R$ is introduced to compare entropy production method and pressure drop method for different components of the pump device at different flow rates, and $R$ is calculated by equation (5.3).

$$R = \frac{h_{pro}}{h_p}. \tag{5.3}$$

Figure 9 is the hydraulic loss ratio $R$ value of different parts of the pump device under various flow conditions. The ratio $R$ of the elbow inlet conduit fluctuates between 0.97 and 1.2, and the ratio $R$ of the straight pipe outlet conduit decreases first and then increases with the increase of the flow rate, ranging from 0.72 to 1.05. The ratio $R$ of impeller domain and guide vane domain is relatively small. The ratio $R$ of impeller domain increases with the increase of flow rate, and the fluctuation range is 0.58–1.04. The ratio $R$ range of guide vane domain is 0.6–0.92. Considering the good consistency of hydraulic loss calculated by entropy production method and pressure drop method in elbow inlet conduit and straight pipe outlet conduit, the reason why the ratio $R$ of impeller domain and guide vane domain is small may be related to the high rotation of the impeller domain and the high-speed flow. After flowing from the impeller domain into the guide vane domain, the fluid still has strong rotational strength and velocity. The follow-up work will focus on the influence of the strong flow rotation intensity in the impeller domain on the hydraulic loss calculated by the entropy production and propose corresponding improvement measures. It is obvious that the ratio $R$ of each component of the pump device is closest to 1 at $1.2Q_{bep}$, the hydraulic loss calculated by entropy production is in good agreement with the hydraulic loss obtained by pressure drop. Meanwhile, the ratio $R$ values of guide vane domain and impeller domain are the highest, 0.92 and 1.04, respectively. On the whole, the results of hydraulic loss calculated by entropy generation are consistent with those calculated by pressure drop method to some extent. Therefore, the entropy generation calculated by the wall equation used in this paper is credible within a certain error range.

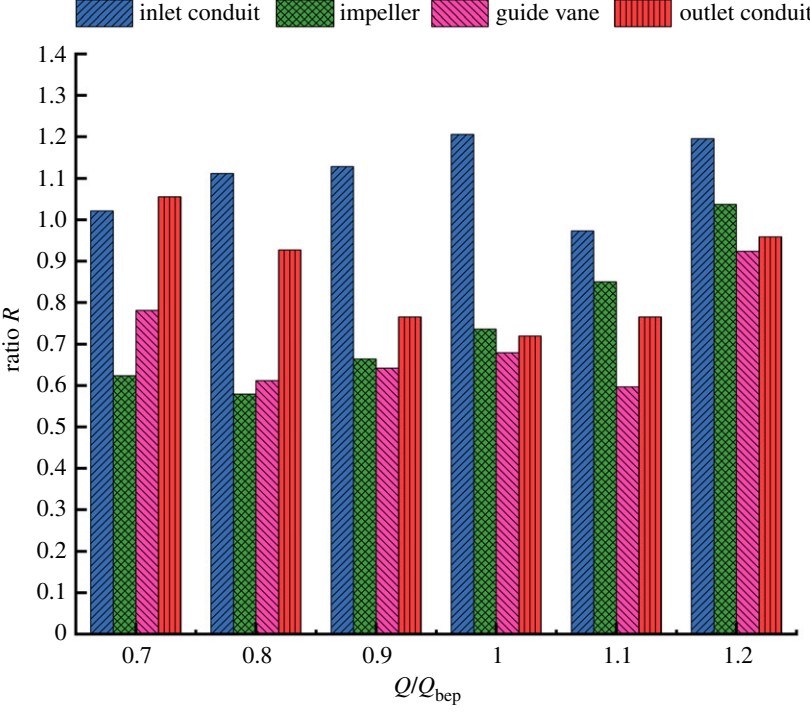

**Figure 9.** Ratio $R$ of hydraulic loss calculated by entropy production to pressure drop under various flow rates.

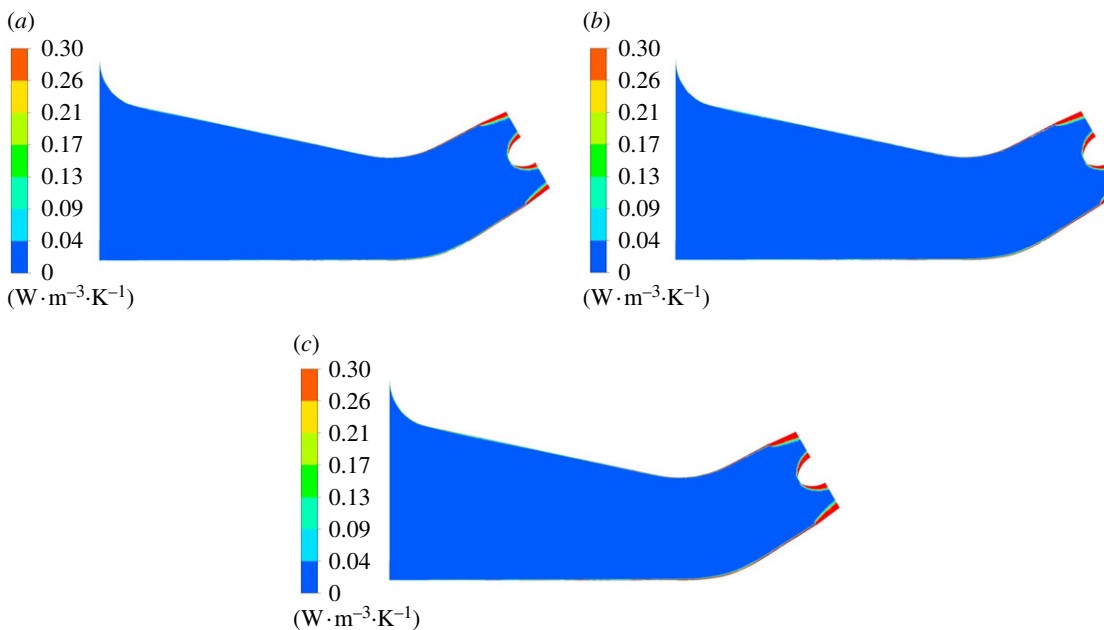

**Figure 10.** Mainstream entropy production rate distribution of elbow inlet conduit. (a) $0.8Q_{bep}$. (b) $1.0Q_{bep}$. (c) $1.2Q_{bep}$.

## 5.4. Detailed entropy distribution rate of pump components

### 5.4.1. Entropy production distribution rate of elbow inlet conduit

Mainstream entropy production distribution in elbow inlet conduit is shown in figure 10. Under various flow rates, the high entropy production rate is concentrated on water cap wall and conduit wall at elbow inlet conduit outlet, and high entropy production rate distribution range increases as the flow rate grows. This is due to the good flow pattern in elbow inlet conduit, and the flow loss is small in the inlet conduit. The flow encounters the water cap at the inlet conduit outlet to generate flow around. With the increase of flow disturbance near the wall, the entropy production rate increases.

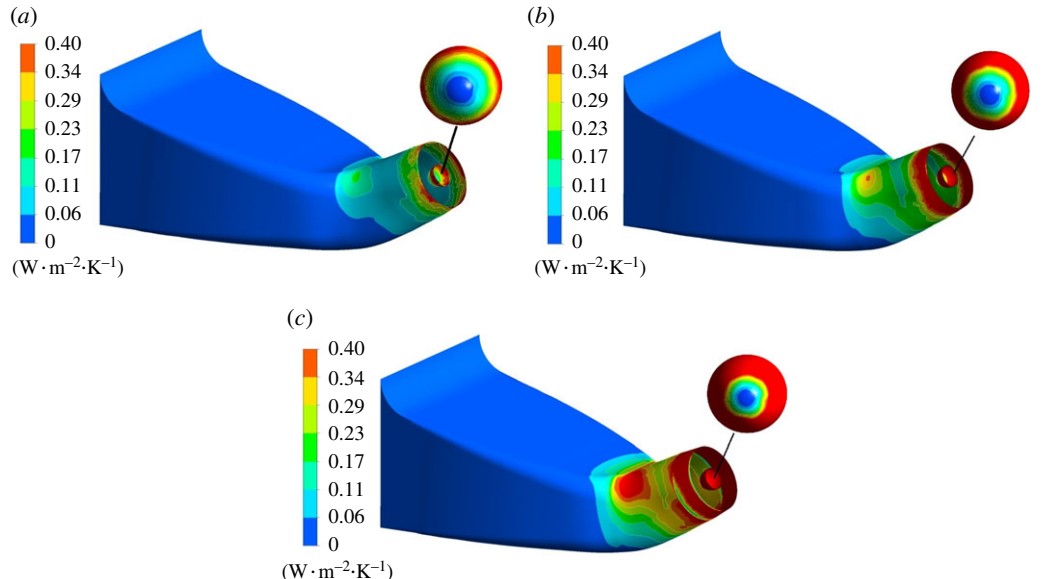

**Figure 11.** Wall entropy production rate distribution of water cap. (a) $0.8Q_{bep}$. (b) $1.0Q_{bep}$. (c) $1.2Q_{bep}$.

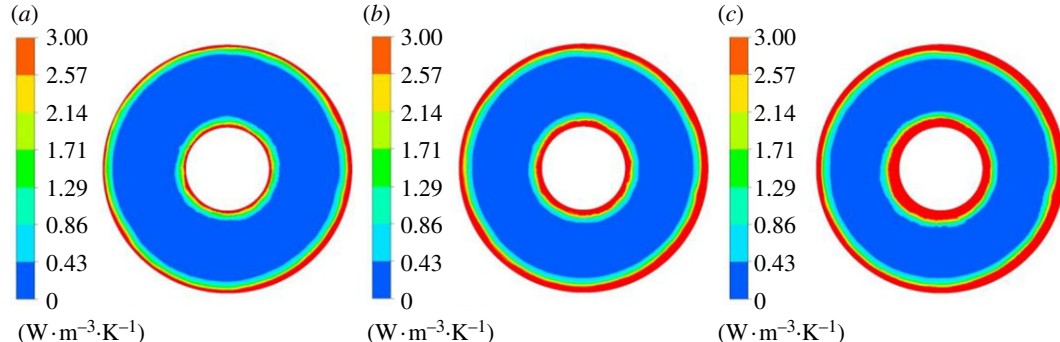

**Figure 12.** Mainstream entropy production rate distribution of the impeller inlet section. (a) $0.8Q_{bep}$. (b) $1.0Q_{bep}$. (c) $1.2Q_{bep}$.

Figure 11 shows the local amplification of water cap wall entropy production rate distribution, which is mainly distributed at elbow section wall and water cap wall in inlet conduit. The elbow section cross-sectional area decreases and flow direction changes to the oblique direction, which causes the increase of flow velocity and more friction of water and wall that causes an increase in entropy production rate. With the influence of centrifugal force, the flow is easy to form outflow at the upper wall of elbow section, and the entropy production loss grows. As the flow rate grows, the entropy production loss at the upper wall grows. The entropy production loss of guide cap top is small, which has a relationship with the deflection of water flow around guide cap and the small flow. As the flow rate grows, high entropy production rate distribution range on water cap wall gradually increases.

### 5.4.2. Entropy production distribution rate of impeller

Figure 12 is the mainstream entropy production rate distribution in impeller inlet section. The mainstream entropy production around the shroud and hub is higher, and the entropy production rate decreases to the conduit centre. The high entropy production rate range grows as the flow rate grows, and the increasing flow velocity leads to the collision between water flow and wall, resulting in the increase of entropy production loss. In mainstream region, the entropy production is small as the good flow pattern.

Figure 13 shows the mainstream entropy production rate of span = 0.2, 0.5 and 0.8 blades under different flow conditions. When flow rate varies, the law of entropy production rate distribution is consistent, and entropy production rate concentrates on the working face of impeller blades approaching impeller outlet. Under $0.8Q_{bep}$ flow condition, the aerofoil structure near the hub affects the flow and collides with the outlet edge of the blade working face, which is easy to generate

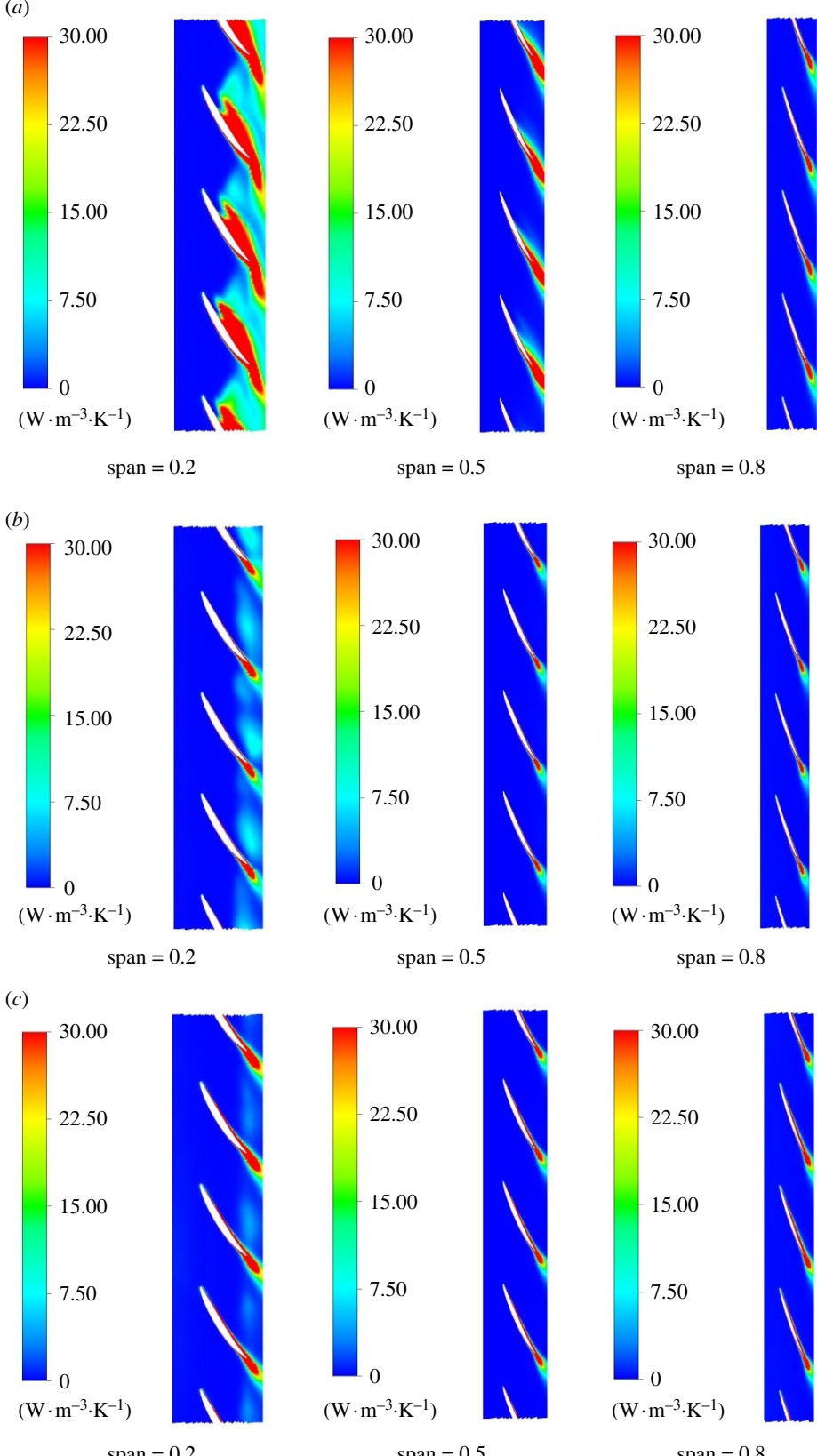

**Figure 13.** Mainstream entropy production rate of blade spanwise at different positions. (*a*) $0.8Q_{bep}$. (*b*) $1.0Q_{bep}$. (*c*) $1.2Q_{bep}$.

outflow and secondary reflux, which increases the entropy production rate. The flow rate has a great effect on entropy production rate. Under $1.0Q_{bep}$ and $1.2Q_{bep}$ flow conditions, the entropy production of the blade working face decreases significantly at the section of span = 0.2 compared with $0.8Q_{bep}$.

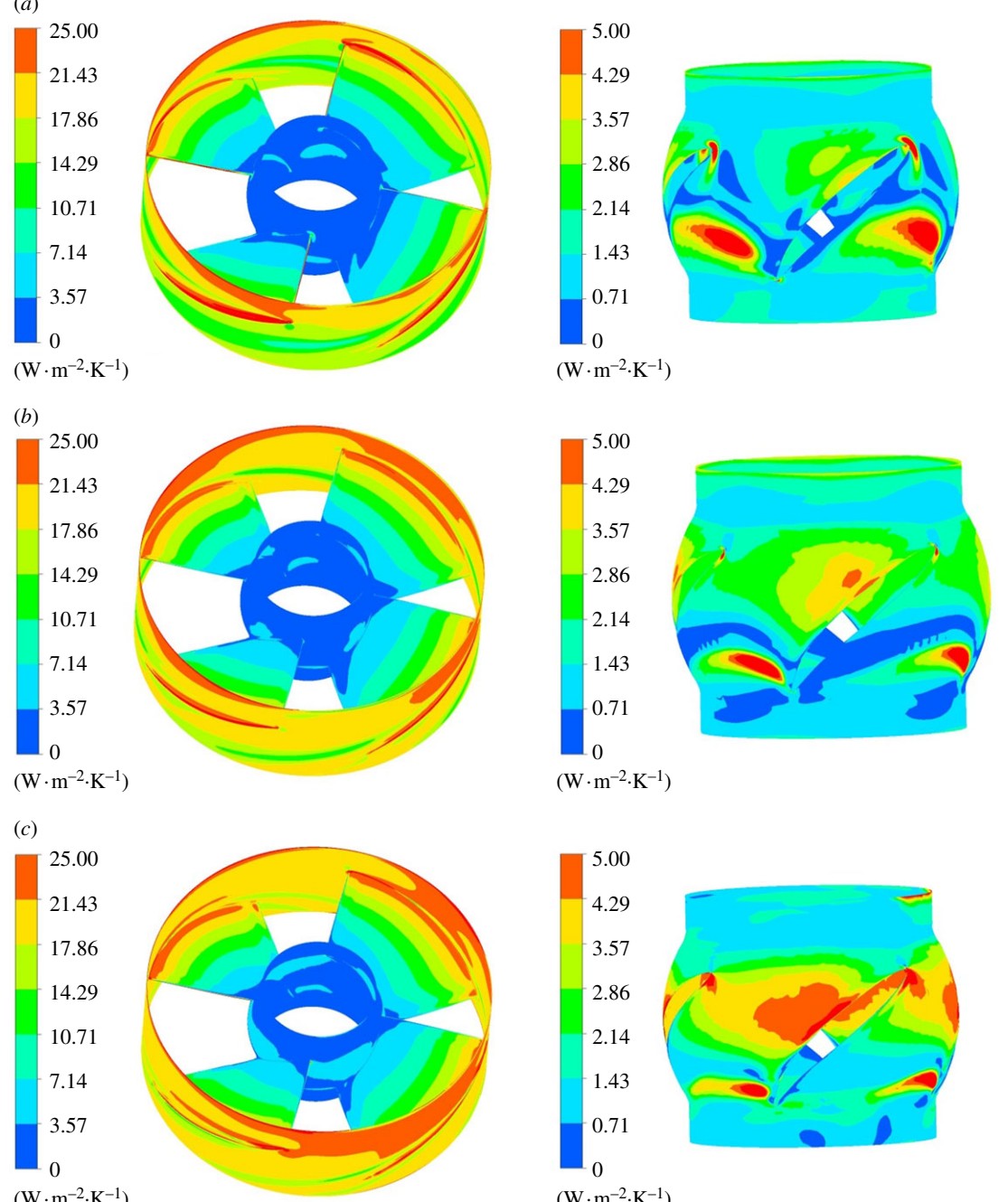

**Figure 14.** Wall entropy production rate distribution in impeller domain. Left: wall entropy production rate of impeller. Right: wall entropy production rate of hub. (a) $0.8Q_{bep}$. (b) $1.0Q_{bep}$. (c) $1.2Q_{bep}$.

The entropy production at the tail of the blade at span = 0.8 is smaller than that at span = 0.2 at different flow conditions, and the blade tail's entropy production is the minimum at $1.0Q_{bep}$.

The entropy production loss on impeller wall is displayed in figure 14. The entropy production of impeller hub wall is significantly lower than that of the blade surface and the rim. The hub is selected for local amplification to analyse the entropy production rate distribution law. When the impeller rotates, the flow will generate centrifugal force, which causes great friction with the pump shell. Therefore, the entropy production rate at impeller wall rim is higher than that at hub, and the entropy production rate decreases from the impeller rim to the hub, showing a gradient distribution. As the blade rotates, the inlet edge of impeller blade will generate local small vortex, and the gap leakage will occur between the blade and the pump shell, which disturbs the flow pattern. The wall entropy production rate is the highest at the tip clearance. When the flow rotates out of the impeller domain, the flow pattern is more complex than the inlet and the entropy production loss is large.

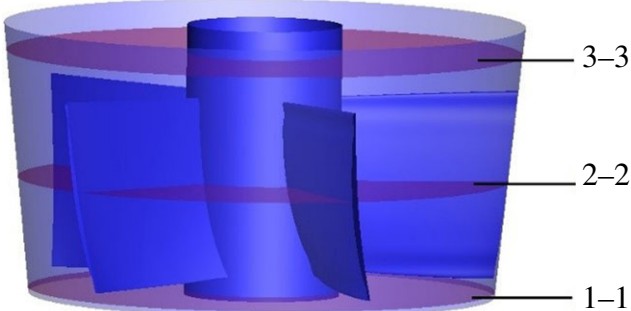

**Figure 15.** Characteristic sections diagram of guide vane domain.

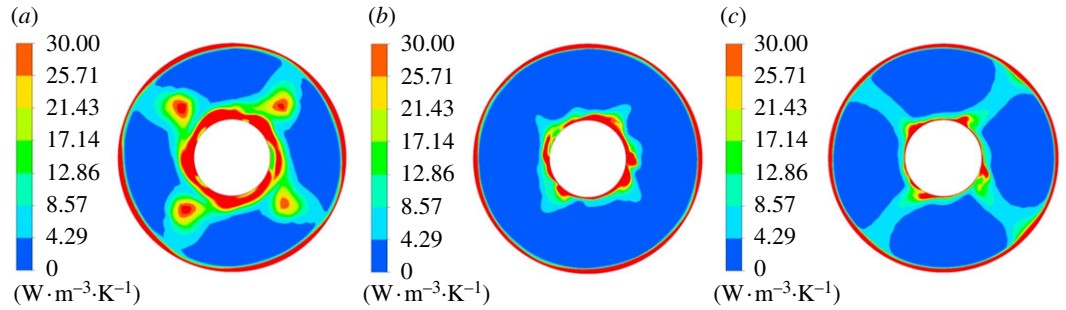

**Figure 16.** Mainstream entropy production rate distribution of section 1–1. (a) $0.8Q_{bep}$. (b) $1.0Q_{bep}$. (c) $1.2Q_{bep}$.

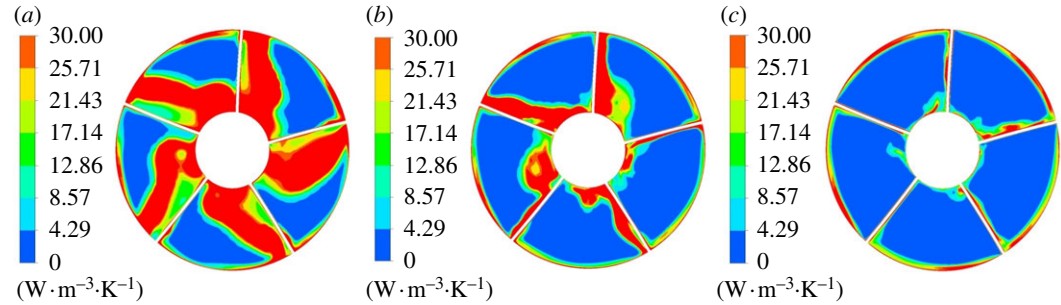

**Figure 17.** Mainstream entropy production rate distribution of section 2–2. (a) $0.8Q_{bep}$. (b) $1.0Q_{bep}$. (c) $1.2Q_{bep}$.

### 5.4.3. Entropy production distribution rate of guide vane

In order to investigate the mainstream entropy production rate distribution of guide vane domain, three characteristic sections of the guide vane are selected for analysis. Figure 15 shows the schematic diagram of the position of the characteristic sections. The characteristic section 1–1 is the guide vane inlet, $0.179D$ away from the impeller centre, section 2–2 is $0.417D$ away from the impeller centre, section 3–3 is $0.690D$ away from the impeller centre.

Figure 16 shows mainstream entropy production rate distribution of section 1–1. Under different flow conditions, there are high entropy production rate regions around the hub and rim. At $0.8Q_{bep}$ flow condition, there are four high entropy production rate regions with equal spacing distribution between the hub and rim, and the entropy production loss around the hub is large and has wide range. Compared with the condition of $0.8Q_{bep}$, the high entropy production rate range around the hub is reduced at $1.0Q_{bep}$ flow condition, and there is no high entropy production rate region in flow passage centre. Under $1.2Q_{bep}$ flow condition, the high entropy production rate distribution range around the guide vane hub further decreases, and there are mainly four regions with small distribution range of high entropy production rate. It indicates that the impeller rotation has a great influence on the entropy production rate distribution at the guide vane inlet under non-design conditions, especially under $0.8Q_{bep}$ conditions.

Figure 17 displays the mainstream entropy production rate distribution of section 2–2. Guide vane blades divide the flow passage into five single passages in guide vane domain. One side of the passage is the pressure surface, whereas the other side is the blade suction surface. The pressure surface's flow velocity is large, and the entropy production rate is small with the good flow pattern. The suction surface flow velocity is small, which causes the larger entropy production rate. Under

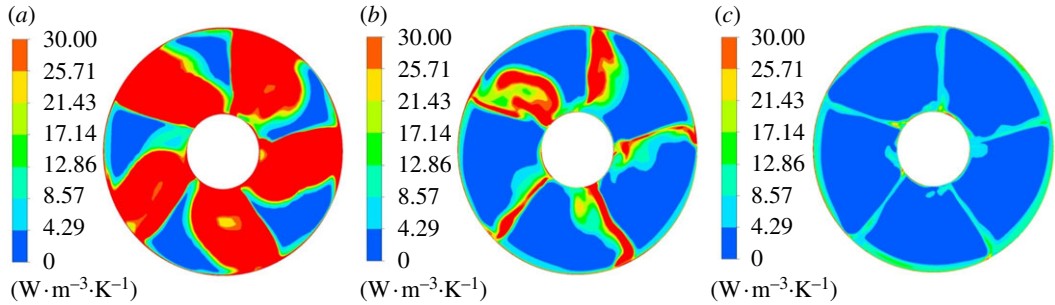

**Figure 18.** Mainstream entropy production rate distribution of section 3–3. (a) $0.8Q_{bep}$. (b) $1.0Q_{bep}$. (c) $1.2Q_{bep}$.

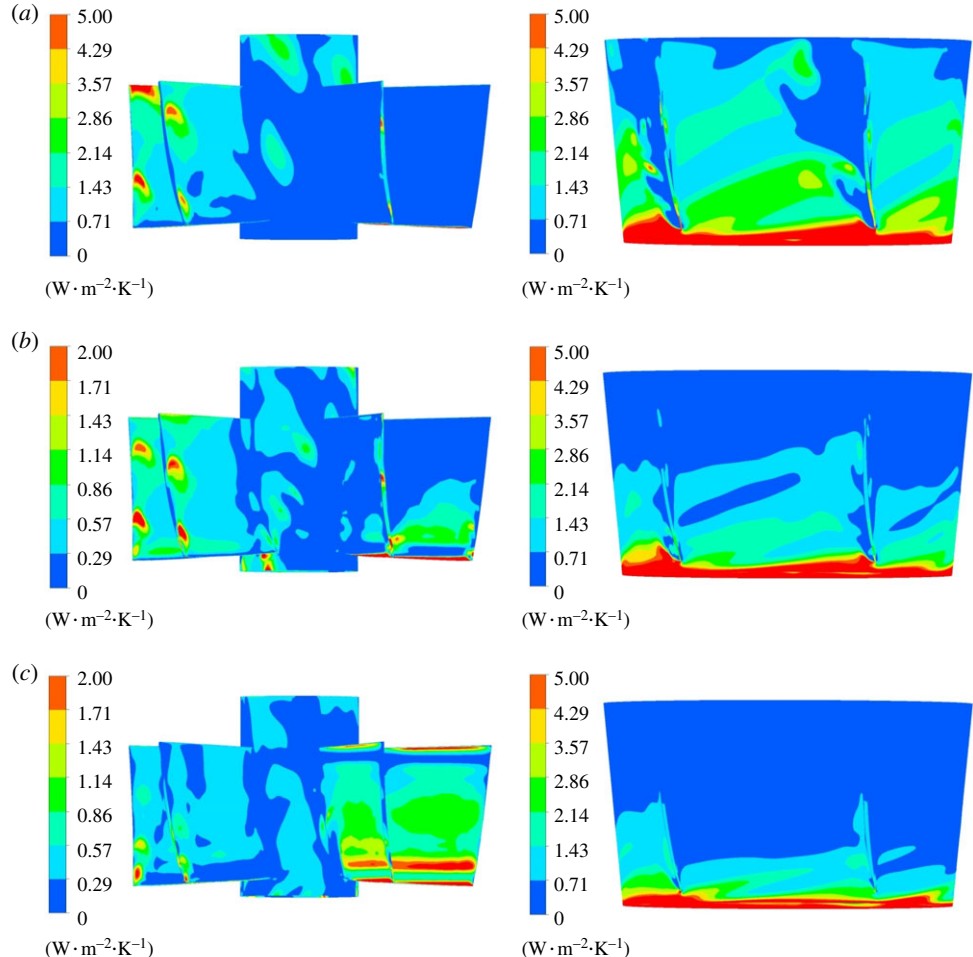

**Figure 19.** Wall entropy production rate distribution of guide vane domain. Left: hub of guide vane. Right: shroud of guide vane. (a) $0.8Q_{bep}$. (b) $1.0Q_{bep}$. (c) $1.2Q_{bep}$.

$0.8Q_{bep}$ flow condition, the flow pattern is poor, and the local flows in the guide vane domain collide with each other, resulting in backflow, which increases the entropy production rate. As the flow rate grows, high entropy production rate region range gradually decreases.

Figure 18 shows the mainstream entropy production rate distribution of section 3–3. The entropy production rate distribution roughly presents five regions with similar distribution rules under different flow conditions. Under $0.8Q_{bep}$ flow condition, the entropy production loss was large, and high entropy production rate range is larger than section 2–2. Entropy production rate range approaching rim increases a lot compared with section 2–2, and the entropy production rate decreased as the flow rate grows. At $1.2Q_{bep}$ flow rate, the entropy production rate was significantly reduced compared with the other two flow conditions.

Figure 19 is the wall entropy production rate distribution in guide vane domain. The intensity of wall entropy production rate is lower than mainstream entropy production rate intensity, indicating that the mainstream entropy production occupies a dominant position in the guide vane domain. There exists

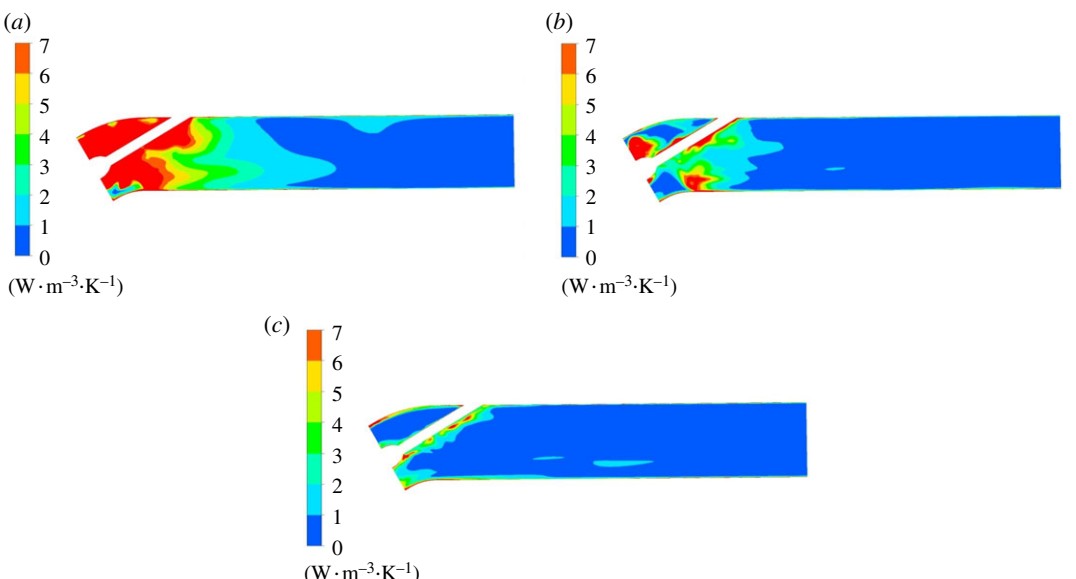

**Figure 20.** Mainstream entropy production rate distribution of outlet conduit. (a) $0.8Q_{bep}$. (b) $1.0Q_{bep}$. (c) $1.2Q_{bep}$.

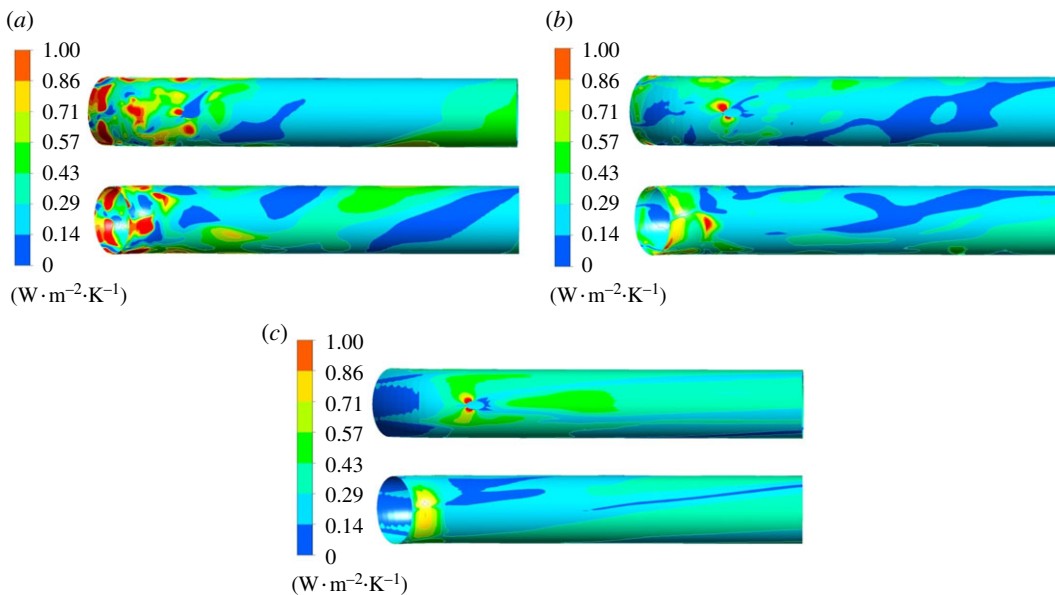

**Figure 21.** Wall entropy production rate distribution of outlet conduit. (a) $0.8Q_{bep}$. (b) $1.0Q_{bep}$. (c) $1.2Q_{bep}$.

high entropy production rate region at the inlet of guide vane shroud, which is obviously affected by dynamic and static interference. As the flow rate grows, the entropy production rate at guide vane inlet shroud gradually decreases. The entropy production rate of hub is small at various working conditions. Under $0.8Q_{bep}$ flow condition, the high entropy production rate region exists on blade's pressure surface. Under $1.0Q_{bep}$ flow condition, the entropy production rate of the pressure surface decreases, and a small range of high entropy production region emerges on blade's suction surface. As the flow rate grows, entropy production rate on guide vane blade's pressure surface further decreases, while with an increased high entropy production rate range on the suction surface.

### 5.4.4. Entropy production distribution rate of straight pipe outlet conduit

Figure 20 is the mainstream entropy production rate distribution of the characteristic section in the straight pipe outlet conduit. The maximum entropy production rate in the straight pipe outlet conduit is obviously lower than entropy production rate in impeller and guide vane domains. In elbow section, the entropy production rate presents a decreasing trend from the elbow section to the conduit outlet. Under $0.8Q_{bep}$ flow condition, the flow formed a bad flow pattern in the elbow section with the influence of centrifugal force and residual circulation, which results in a high entropy production rate area with a

large gradient. Under $1.0Q_{bep}$ flow condition, the flow pattern is improved, the high entropy production rate range is greatly decreased. There exists a part high entropy production rate region.

Figure 21 is wall entropy production rate distribution on the top and bottom of the straight pipe outlet conduit. The wall entropy production rate of outlet conduit is obviously lower than that of mainstream. At $0.8Q_{bep}$ flow condition, the entropy production rate at the outlet conduit wall is mainly distributed in the elbow section, and entropy production rate distribution is disordered, which is due to flow direction change and poor flow pattern in elbow section. As the flow rate grows, the entropy production rate of elbow section decreases obviously. The spiral high entropy production rate area in the straight pipe section of the outlet conduit is distributed on the straight section. When the flow rate increases, the flow pattern improves and straight section's entropy production rate in outlet conduit decreases.

# 6. Conclusion

Numerical simulation reliability is proved by comparing the results of model test and numerical simulation in this paper. The entropy production theory analysis of components in whole 30° slanted axial-flow pump device is carried out, and the conclusions are drawn as follows:

The entropy production in impeller domain is the highest among the pump components. When the flow rate grows, entropy production proportion of the impeller domain in pump device total entropy production increases continuously, which is related to the complex flow pattern inside impeller domain. The entropy production of straight pipe outlet conduit is large at small flow rate. As the flow rate grows, the entropy production of straight pipe outlet conduit decreases sharply. The variation trend of entropy production in guide vane domain with various flow rates is similar to that of straight pipe outlet conduit, but the entropy production is smaller as a whole. The flow pattern inside the elbow inlet conduit is good, the entropy production in elbow inlet conduit accounts for a very small proportion in whole flow conduit's total entropy production.

Under small flow conditions, turbulent dissipation entropy production's proportion and wall dissipation entropy production in total entropy production is much higher than direct dissipation entropy production. As the flow rate grows, the proportion of direct dissipation entropy production decreases, and wall dissipation entropy production proportion increases. At $0.7Q_{bep}$, the proportion of direct dissipation entropy production is close to 74%, which is about 2.8 times the wall dissipation entropy production. Under $1.2Q_{bep}$ flow condition, the proportion of direct dissipation entropy production is only 5.5% higher than that of wall dissipation entropy production.

The overall flow pattern of elbow inlet conduit is good with the low entropy production rate. The mainstream entropy production rate and wall entropy production rate are concentrated in outlet section of inlet conduit. The wall entropy production rate of impeller, guide vane and outlet conduit are lower than the mainstream entropy production rate, and the mainstream entropy production rate occupies the dominant position. The mainstream entropy production rate of the impeller inlet section is concentrated around the hub and rim. Due to the influence of local outflow caused by aerofoil flow, there exists amount of flow loss at the blade tail. The impeller shroud wall entropy production is mainly caused by tip clearance leakage. The mainstream entropy production of guide vane domain has a wide distribution range in section 3–3, forming five large-scale high entropy production rate regions with similar shapes. The overall entropy production of the outlet conduit is small, and the entropy production rate is basically concentrated in the elbow section. When the flow rate grows, flow pattern is improved so that entropy production rate is reduced in outlet conduit.

Data accessibility. Pump device and entropy production data files are available from the Dryad Digital Repository: https://doi.org/10.5061/dryad.0vt4b8h0g [38].

Authors' contributions. F.Y.: conceptualization, methodology and funding; Z.L.: software, writing—original draft and formal analysis; W.H.: formal analysis and investigation; C.L.: resources and supervision; D.J.: validation; D.L.: visualization; A.N.: writing—review and editing. All authors have read and agreed to the published version of the manuscript.

All authors gave final approval for publication and agreed to be held accountable for the work performed therein.

Competing interests. The authors declare no completing interests.

Funding. This research work was supported by the National Natural Science Foundation of China (grant no. 51609210, 51779214), major projects of the Natural Science Foundation of the Jiangsu Higher Education Institutions of China (grant no. 20KJA570001), the open research subject of Key Laboratory of Fluid and Power Machinery, Ministry of Education (grant no. szjj2016-078), the Science and Technology Plan Project of Yangzhou City (grant no. YZU201901), Technology Project of Water Resources Department of Jiangsu Province (grant no. 2020029), Priority Academic Program Development of Jiangsu Higher Education Institutions [PAPD] and Postgraduate Research & Practice Innovation Program of Jiangsu Province [SJCX21_1583].

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
