## [Peer Review File · Royal Society Open Science]

Review History

RSOS-211208.R0 (Original submission)

Review form: Reviewer 1

Is the manuscript scientifically sound in its present form?

Yes

Are the interpretations and conclusions justified by the results?

Yes

Is the language acceptable?

Yes

Do you have any ethical concerns with this paper?

No

Have you any concerns about statistical analyses in this paper?

No

Recommendation?

Accept with minor revision (please list in comments)

Comments to the Author(s)

This paper introduces some interesting analyses about internal flow loss characteristics of a slanted axial-flow pump based on entropy production theory. The causes of flow loss in each flow component of the slanted axial-flow pump device is revealed, and the entropy production rate distribution cloud is fully displayed. This paper provides a new idea for analyzing the hydraulic loss of the pump device, and the conclusion is beneficial to the hydraulic optimization of axial-flow pump device. Overall, the entire paper is well written and the results have been presented in an organized manner. There are some comments showing as follows,

1. The introduction part should clarify the necessity of using entropy production theory to analyze the flow loss of each component of the pump device, and explain the difference between the traditional method and entropy production theory in hydraulic loss analysis. Moreover, it is necessary to clarify the motivation of this study .
2. The entropy production is quite useful not only to the field of different operating condition, but also to the transient process of pumps. Please pay attention to these two references, Transient characteristics during power-off process in a shaft extension tubular pump by using a suitable numerical model, Numerical simulation of transient flow in a shaft extension tubular pump unit during runaway process caused by power failure.
3. The conversion principles and methods of the energy performance of the prototype and model pump devices should be completed in subsection 5.1.
4. The RNG k- ϵ turbulence model was selected for simulations and the corresponding reasons and literature support were given. It is recommended to further explain why this turbulence model is selected or its applicability compared with other turbulence models.
5. Replace "blades" by "blade" on page 5, line 38.
6. There are a few grammatical errors in the manuscript and check carefully.
7. The flow loss mechanism is also relevant to turbulent kinetic energy transport, the authors may enhance the analyses in the view of tke by referring to the following article, numerical study of turbulent flow past a rotating axial-flow pump based on a level-set immersed boundary method.

Review form: Reviewer 2

Is the manuscript scientifically sound in its present form?

No

Are the interpretations and conclusions justified by the results?

No

Is the language acceptable?

Yes

Do you have any ethical concerns with this paper?

No

Have you any concerns about statistical analyses in this paper?

No

Recommendation?

Major revision is needed (please make suggestions in comments)

Comments to the Author(s)

See attached file (Appendix A).

Decision letter (RSOS-211208.R0)

Dear Professor Yang

The Editors assigned to your paper RSOS-211208 "Analysis of the flow loss characteristics of a slanted axial-flow pump device based on entropy production theory" have now received comments from reviewers and would like you to revise the paper in accordance with the reviewer comments and any comments from the Editors. Please note this decision does not guarantee eventual acceptance.

Please submit your revised manuscript and required files (see below) no later than 21 days from today's (ie 29-Oct-2021) date. Note: the ScholarOne system will 'lock' if submission of the revision is attempted 21 or more days after the deadline. If you do not think you will be able to meet this deadline please contact the editorial office immediately.

on behalf of Professor R. Kerry Rowe (Subject Editor)
openscience@royalsociety.org

Associate Editor Comments to Author:

Although the referees identify a number of aspects of the work that may make it eventually suitable for publication, there are a number of concerns that mean that, at present, we're unable to recommend acceptance. Please carefully review the queries and comments of the referees and prepare a revision that addresses the comments made - you should make it clear in a tracked-changes paper and point-by-point response document when you resubmit. The decision email will provide guidance from the editorial office regarding the preparation of a revision.

Reviewer comments to Author:

Reviewer: 1

Comments to the Author(s)

This paper introduces some interesting analyses about internal flow loss characteristics of a slanted axial-flow pump based on entropy production theory. The causes of flow loss in each flow component of the slanted axial-flow pump device is revealed, and the entropy production rate distribution cloud is fully displayed. This paper provides a new idea for analyzing the hydraulic loss of the pump device, and the conclusion is beneficial to the hydraulic optimization of axial-flow pump device. Overall, the entire paper is well written and the results have been presented in an organized manner. There are some comments showing as follows,

1. The introduction part should clarify the necessity of using entropy production theory to analyze the flow loss of each component of the pump device, and explain the difference between the traditional method and entropy production theory in hydraulic loss analysis. Moreover, it is necessary to clarify the motivation of this study .
2. The entropy production is quite useful not only to the field of different operating condition, but also to the transient process of pumps. Please pay attention to these two references, Transient characteristics during power-off process in a shaft extension tubular pump by using a suitable numerical model, Numerical simulation of transient flow in a shaft extension tubular pump unit during runaway process caused by power failure.
3. The conversion principles and methods of the energy performance of the prototype and model pump devices should be completed in subsection 5.1.
4. The RNG $k-\epsilon$ turbulence model was selected for simulations and the corresponding reasons and literature support were given. It is recommended to further explain why this turbulence model is selected or its applicability compared with other turbulence models.
5. Replace "blades" by "blade" on page 5, line 38.
6. There are a few grammatical errors in the manuscript and check carefully.
7. The flow loss mechanism is also relevant to turbulent kinetic energy transport, the authors may enhance the analyses in the view of tke by referring to the following article, numerical study of turbulent flow past a rotating axial-flow pump based on a level-set immersed boundary method.

Reviewer: 2

Comments to the Author(s)

See attached file ("review.pdf").

===PREPARING YOUR MANUSCRIPT===

If you have been asked to revise the written English in your submission as a condition of publication, you must do so, and you are expected to provide evidence that you have received language editing support. The journal would prefer that you use a professional language editing service and provide a certificate of editing, but a signed letter from a colleague who is a fluent speaker of English is acceptable. Note the journal has arranged a number of discounts for authors using professional language editing services (<https://royalsociety.org/journals/authors/benefits/language-editing/>).

===PREPARING YOUR REVISION IN SCHOLARONE===

- If you are providing image files for potential cover images, please upload these at this step, and inform the editorial office you have done so. You must hold the copyright to any image provided.
- A copy of your point-by-point response to referees and Editors. This will expedite the preparation of your proof.

- Ensure that your data access statement meets the requirements at <https://royalsociety.org/journals/authors/author-guidelines/#data>. You should ensure that you cite the dataset in your reference list. If you have deposited data etc in the Dryad repository, please include both the 'For publication' link and 'For review' link at this stage.
- If you are requesting an article processing charge waiver, you must select the relevant waiver option (if requesting a discretionary waiver, the form should have been uploaded at Step 3 'File upload' above).
- If you have uploaded ESM files, please ensure you follow the guidance at <https://royalsociety.org/journals/authors/author-guidelines/#supplementary-material> to include a suitable title and informative caption. An example of appropriate titling and captioning may be found at https://figshare.com/articles/Table_S2_from_Is_there_a_trade-off_between_peak_performance_and_performance_breadth_across_temperatures_for_aerobic_scop_e_in_teleost_fishes_/3843624.

Author's Response to Decision Letter for (RSOS-211208.R0)

See Appendix B.

RSOS-211208.R1 (Revision)

Review form: Reviewer 1

Is the manuscript scientifically sound in its present form?

Yes

Are the interpretations and conclusions justified by the results?

Yes

Is the language acceptable?

Yes

Do you have any ethical concerns with this paper?

No

Have you any concerns about statistical analyses in this paper?

No

Recommendation?

Accept as is

Comments to the Author(s)

Interesting and nice work ! Good luck !

Review form: Reviewer 2**Is the manuscript scientifically sound in its present form?**

Yes

Are the interpretations and conclusions justified by the results?

Yes

Is the language acceptable?

Yes

Do you have any ethical concerns with this paper?

No

Have you any concerns about statistical analyses in this paper?

No

Recommendation?

Accept as is

Comments to the Author(s)

Thank you for revising the paper.

Decision letter (RSOS-211208.R1)

Dear Professor Yang,

It is a pleasure to accept your manuscript entitled "Analysis of the flow loss characteristics of a slanted axial-flow pump device based on entropy production theory" in its current form for publication in Royal Society Open Science. The comments of the reviewer(s) who reviewed your manuscript are included at the foot of this letter.

Please ensure that you send to the editorial office an editable version of your accepted manuscript, and individual files for each figure and table included in your manuscript. You can send these in a zip folder if more convenient. Failure to provide these files may delay the processing of your proof.

on behalf of Prof R. Kerry Rowe (Subject Editor)
openscience@royalsociety.org

Associate Editor Comments to Author:

The reviewers agree that the paper is now ready for acceptance after revision. Thank you for supporting the journal.

Reviewer comments to Author:

Reviewer: 1
Comments to the Author(s)
Interesting and nice work ! Good luck !

Reviewer: 2
Comments to the Author(s)
Thank you for revising the paper.

Appendix A

The paper presents a loss analysis by entropy production in an axial flow pump. The losses distribution within the flow field and at the walls is assessed. I have some remarks / questions.

1. Section 2.2: the authors state a y^+ value in the range of logarithmic wall functions. Why has no low-Reynolds approach been tested?
2. Section 2.3: steady approach: not state of the art. "Stage" and "none" approach are an internal term of the software vendor. Here, the algorithms should be described.
3. It can be concluded from section 2.5 that the authors solve the RANS equations. This is however not explicitly stated.
4. In Eq 13, the near-wall approach is described. In recent literature e.g. [8,12,19], it is pointed out that a considerable error is made in wall proximity when the grid resolution is in the range of a logarithmic law, and if no entropy wall function is used. Note also reference [x] below. Although the authors use a rather coarse near-wall resolution, they ignore this important point, which is my main criticism on this work. It is indispensable that the authors reflect their procedure in terms of Eq 13 to the more advanced near-wall treatment of relevant literature.

[X] Kock, F., and Herwig, H., 2004, "Local Entropy Production in Turbulent Shear Flows: A high-Reynolds Number Model With Wall Functions," *Int. J. Heat Mass Transfer*, 47(10–11), pp. 2205–2215.

5. For example, in the abstract and in the conclusions the authors assess the wall entropy production. However, it is not clear how the simplified wall treatment affects wall entropy production. See also point 4 above.

Assessment

The paper is well written and easy to read. Generally, the task of loss evaluation in circumferential pumps is relevant, albeit the authors use quite standard CFD methods which do not contain any novelty. A novelty might be the application of the entropy production method to this particular pump test case. However, I have a significant concern regarding its reliability in wall proximity as stated in point 4 and 5 above. This needs to be rectified, before I can recommend the paper for publication. Beyond providing a reasoning for their particular wall treatment, the authors may want to provide a comparison between volume-integral losses by entropy production with the total pressure loss of a non-rotating component, for example a channel flow. Both should coincide if the wall treatment is appropriate.

Appendix B

Associate Editor Comments to Author:

Although the referees identify a number of aspects of the work that may make it eventually suitable for publication, there are a number of concerns that mean that, at present, we're unable to recommend acceptance. Please carefully review the queries and comments of the referees and prepare a revision that addresses the comments made - you should make it clear in a tracked-changes paper and point-by-point response document when you resubmit. The decision email will provide guidance from the editorial office regarding the preparation of a revision.

Response to editor: Thank you for your letter and for the reviewers' comments concerning our manuscript entitled "Analysis of flow loss characteristics of slanted axial-flow pump device based on entropy production theory" (ID: RSOS-211208). Those comments are all valuable and very helpful for revising and improving our paper, as well as the important guiding significance to our researches. We have studied comments carefully and have made correction which we hope meet with approval. Revised portion are marked by tracked changes in the paper. The main corrections in the paper and the responds to the reviewer's comments are as flowing:

Reply to Reviewer #1

Comments:

This paper introduces some interesting analyses about internal flow loss characteristics of a slanted axial-flow pump based on entropy production theory. The causes of flow loss in each flow component of the slanted axial-flow pump device is revealed, and the entropy production rate distribution cloud is fully displayed. This paper provides a new idea for analyzing the hydraulic loss of the pump device, and the conclusion is beneficial to the hydraulic optimization of axial-flow pump device. Overall, the entire paper is well written and the results have been presented in an organized manner. There are some comments showing as follows,

We appreciate your clear and detailed feedback and hope that the explanation has fully addressed all of your concerns. In the remainder of this letter, we discuss each of your comments individually along with our corresponding responses.

To facilitate this discussion, we first retype your comments in *italic* font and then present our responses to the comments in red.

1. The introduction part should clarify the necessity of using entropy production theory to analyze the flow loss of each component of the pump device, and explain the difference between the traditional method and entropy production theory in hydraulic loss analysis. Moreover, it is necessary to clarify the motivation of this study.

Response 1: The traditional method of estimating hydraulic loss is to calculate the pressure drop on the inlet and outlet surface to obtain the relatively general hydraulic loss in the channel, while the detailed distribution and size of hydraulic loss at different positions in the channel cannot be known. The entropy production theory has obvious advantages in the evaluation of hydraulic loss, which can accurately locate the source of hydraulic loss, and quantitatively and qualitatively analyze the size and distribution of hydraulic loss. Therefore, it can provide a theoretical basis for the improvement and

optimization of components with large hydraulic loss.

2. *The entropy production is quite useful not only to the field of different operating condition, but also to the transient process of pumps. Please pay attention to these two references, Transient characteristics during power-off process in a shaft extension tubular pump by using a suitable numerical model, Numerical simulation of transient flow in a shaft extension tubular pump unit during runaway process caused by power failure.*

Response 2: Thanks to the reviewer's suggestions, the entropy production theory has a broad application prospect, which can be used to evaluate the hydraulic loss in different types and forms of water conservancy projects operating in different conditions. The follow-up work will pay more attention to the transient flow process of the flow in the slanted axial flow pump device under the bad operation condition or power-off process of the pump station.

3. *The conversion principles and methods of the energy performance of the prototype and model pump devices should be completed in subsection 5.1.*

Response 3: Due to the great difference in size between prototype pump and model pump, in order to ensure the correlation of energy performance between prototype pump and model pump, the size conversion between prototype pump device and model pump device is carried out according to the principle of equal nD value converted by equal head of similarity law.

4. *The RNG k - ε turbulence model was selected for simulations and the corresponding reasons and literature support were given. It is recommended to further explain why this turbulence model is selected or its applicability compared with other turbulence models.*

Response 4: The RNG k - ε turbulence model is based on the renormalization group analysis of the N-S equation. Compared with the standard k - ε turbulence model, the model constants of the turbulent kinetic energy dissipation rate ε transport equation is different, and the calculation accuracy of the RNG k - ε turbulence model for the turbulent dissipation term in the flow separation region is improved. This correction takes into account the rotation effect in the average flow and has advantages in dealing with large curvature, strong rotation and high strain rate flow in the pump impeller.

5. *Replace "blades" by "blade" on page 5, line 38.*

Response 5: Modification has been made in the corresponding location.

6. *There are a few grammatical errors in the manuscript and check carefully.*

Response 6: Authors checked the full text and corrected several grammatical errors.

7. *The flow loss mechanism is also relevant to turbulent kinetic energy transport, the authors may enhance the analyses in the view of tke by referring to the following article, numerical study of turbulent flow past a rotating axial-flow pump based on a level-set immersed boundary method.*

Response 7: Thanks to the reviewer's advice, this paper uses entropy production theory to evaluate the flow loss of pump operating conditions to reveal the size and distribution of flow loss. Subsequent studies will focus on the relationship between flow loss and TKE.

-----End of Reply to Reviewer #1-----

Reply to Reviewer #2

Comments:

The paper presents a loss analysis by entropy production in an axial flow pump. The losses distribution within the flow field and at the walls is assessed. I have some remarks / questions.

The paper is well written and easy to read. Generally, the task of loss evaluation in circumferential pumps is relevant, albeit the authors use quite standard CFD methods which do not contain any novelty. A novelty might be the application of the entropy production method to this particular pump test case. However, I have a significant concern regarding its reliability in wall proximity as stated in point 4 and 5 above. This needs to be rectified, before I can recommend the paper for publication. Beyond providing a reasoning for their particular wall treatment, the authors may want to provide a comparison between volume-integral losses by entropy production with the total pressure loss of a non-rotating component, for example a channel flow. Both should coincide if the wall treatment is appropriate.

We appreciate your clear and detailed feedback. You have doubts about the reliability of the calculation results of wall entropy production in the article. We fully understand and treat this point carefully. Authors add content to compare the hydraulic loss calculated by entropy generation and pressure drop to prove the reliability of the wall entropy production calculation method, and we hope that the explanation has fully addressed all of your concerns. In the remainder of this letter, we discuss each of your comments individually along with our corresponding responses.

To facilitate this discussion, we first retype your comments in *italic* font and then present our responses to the comments in red.

1. Section 2.2: the authors state a y^+ value in the range of logarithmic wall functions. Why has no low-Reynolds approach been tested?

Response 1: In the flow channel of pumping station, the flow near the wall belongs to low Reynolds number flow, and the flow away from the wall belongs to high Reynolds number flow. In this paper, the scalable wall function is applied to deal with the near-wall region. As a common method in rotating machinery field [1-3], the scalable wall function can always call the logarithmic law formula to calculate the average velocity without problems when the grid is extremely dense. Limited by the time and test equipment, the authors did not test the low-Reynolds approach. However, the comparison of pump device performance characteristics between test and numerical simulation as well as the verification of the entropy production loss show that the results of numerical simulation are credible within a certain error range, which locate in Section 5.1 and Section 5.3.

[1] Bellary, Sayed Ahmed Imran, and Abdus Samad. "Exit blade angle and roughness effect on centrifugal pump performance." Gas Turbine India Conference. Vol. 35161. American Society of Mechanical Engineers, 2013.

[2] Shukla, S. N., and J. T. Kshirsagar. "Numerical Simulation of Drawdown in Pump Sumps." Fluid Machinery and Fluid Mechanics. Springer, Berlin, Heidelberg, 2009.

352-356.

- [3] He, M. H., et al. "Numerical analysis of an axial-flow pump with different bell mouths." IOP Conference Series: Materials Science and Engineering. Vol. 52. No. 3. IOP Publishing, 2013.

2. *Section 2.3: steady approach: not state of the art. "Stage" and "none" approach are an internal term of the software vendor. Here, the algorithms should be described.*

Response 2: Thanks to reviewer's advice, the explanation of the "Stage" and "None" algorithms will be added in Section 2.3. Stage model refers to the mixing plane model (MPM). MPM performs circumferential average of physical quantities (pressure, velocity, etc.) on the mixing surface between the rotating domain and the stationary domain. And the distribution of physical quantities is introduced into the adjacent computational domain as boundary conditions. With the iteration of the calculation, the distribution of physical quantities on the adjacent boundaries of the mixing surface tends to be consistent until the calculation converges. MPM is mostly applied in hydraulic machinery interface such as axial flow pump and axial flow turbine. None model is suitable for the domains with no frame change or pitch change.

3. *It can be concluded from section 2.5 that the authors solve the RANS equations. This is however not explicitly stated.*

Response 3: The explanation of solving the RANS equation is detailed in Section 2.5. The control equation is discretized by Control Volume Finite Element Method (CV/FEM). The discrete equation is solved by the fully implicit coupled algebraic multi-grid method. In the discretization process, the convection term adopts the high resolution scheme, and the other terms adopt the central difference scheme.

4. *In Eq 13, the near-wall approach is described. In recent literature e.g. [8,12,19], it is pointed out that a considerable error is made in wall proximity when the grid resolution is in the range of a logarithmic law, and if no entropy wall function is used. Note also reference [x] below. Although the authors use a rather coarse near-wall resolution, they ignore this important point, which is my main criticism on this work. It is indispensable that the authors reflect their procedure in terms of Eq 13 to the more advanced near-wall treatment of relevant literature.*

[X] Kock, F., and Herwig, H., 2004, "Local Entropy Production in Turbulent Shear Flows: A high Reynolds Number Model With Wall Functions," *Int. J. Heat Mass Transfer*, 47(10–11), pp. 2205–2215.

5. *For example, in the abstract and in the conclusions the authors assess the wall entropy production. However, it is not clear how the simplified wall treatment affects wall entropy production. See also point 4 above.*

Response 4&5: In the process of fluid transition from the turbulent core far away from the near wall to the laminar boundary layer, obvious velocity gradient will appear near the wall due to the influence of flow viscosity. Kock and Herwig [14] found that using DNS method to calculate entropy production without special treatment of wall entropy production would result in unacceptable error. Zhang et al [34] and Duan et al [35-36] proposed to calculate the entropy production rate in wall region by using the wall shear stress and the velocity near the wall. The entropy production rate in wall region can be calculated in Eq. (13). Scholars [15,16,18,19] verified the reliability of this method by

comparing the hydraulic loss calculated by the entropy production and pressure drop. See Section 3. Entropy production theory in revised version. Therefore, authors add the comparison between the hydraulic loss calculated by comparing the entropy production and pressure drop, as shown in Section 5.3 Validation of hydraulic loss calculated by entropy production.

-----End of Reply to Reviewer #2-----